# Machine Learning for Human Activity Recognition: State-of-the-Art Techniques and Emerging Trends

**DOI:** 10.3390/jimaging11030091

**Published:** 2025-03-20

**Authors:** Md Amran Hossen, Pg Emeroylariffion Abas

**Affiliations:** Faculty of Integrated Technologies, Universiti Brunei Darussalam, Bandar Seri Begawan BE 1410, Brunei; 20h8453@ubd.edu.bn

**Keywords:** activity discovery, activity recognition, sensor modalities, machine learning, deep learning, data fusion, emerging applications

## Abstract

Human activity recognition (HAR) has emerged as a transformative field with widespread applications, leveraging diverse sensor modalities to accurately identify and classify human activities. This paper provides a comprehensive review of HAR techniques, focusing on the integration of sensor-based, vision-based, and hybrid methodologies. It explores the strengths and limitations of commonly used modalities, such as RGB images/videos, depth sensors, motion capture systems, wearable devices, and emerging technologies like radar and Wi-Fi channel state information. The review also discusses traditional machine learning approaches, including supervised and unsupervised learning, alongside cutting-edge advancements in deep learning, such as convolutional and recurrent neural networks, attention mechanisms, and reinforcement learning frameworks. Despite significant progress, HAR still faces critical challenges, including handling environmental variability, ensuring model interpretability, and achieving high recognition accuracy in complex, real-world scenarios. Future research directions emphasise the need for improved multimodal sensor fusion, adaptive and personalised models, and the integration of edge computing for real-time analysis. Additionally, addressing ethical considerations, such as privacy and algorithmic fairness, remains a priority as HAR systems become more pervasive. This study highlights the evolving landscape of HAR and outlines strategies for future advancements that can enhance the reliability and applicability of HAR technologies in diverse domains.

## 1. Introduction

Human activity recognition (HAR) has gained significant attention due to its potential in interpreting human actions across various domains. Consider a high-risk industrial setting, such as an offshore platform, where HAR is used to identify hazardous activities. The effectiveness of HAR in such environments is influenced by external factors, including fluctuating lighting conditions due to changing weather. Vision-based HAR systems may struggle in low-light conditions, potentially missing critical safety incidents. In contrast, depth sensors, which are not affected by lighting conditions, can effectively recognise hazardous movements, allowing real-time safety monitoring and intervention. This example underscores the practical significance of HAR, demonstrating its capability beyond theoretical research.

Beyond workplace safety, HAR plays a crucial role in healthcare, smart environments, sports analytics, surveillance and security systems, autonomous driving systems, human–computer interaction (HCI), and human–robot interaction (HRI) [1,2,3,4,5,6,7]. In healthcare, HAR enables remote patient monitoring [8,9,10], while in smart environments, it has the potential to optimise energy consumption [11]. In sport analytics, HAR enhances sports performance analysis [12], and in security surveillance, it assists in detecting suspicious behaviour [13]. These diverse applications highlight the growing impact of HAR across multiple fields.

Recent advancements in machine learning (ML) and deep learning (DL) technologies [14,15,16,17,18,19,20,21,22,23,24,25] have significantly enhanced the scalability and accuracy of HAR systems [26]. Early stages of HAR research relied primarily on grayscale or RGB videos [27,28], but the field has since expanded over time to include a variety of other sensing modalities, including motion capture devices (Mocap), infrared sequences, event streams, depth sensors, point clouds, 2D/3D skeleton, audio streams, Wi-Fi signals, radio waves, Light Detection and Ranging (LiDAR), and a combination of environmental and proximity sensors [29,30,31,32,33,34,35,36,37,38,39]. The integration of wearable devices such as accelerometers [40,41,42], gyroscopes, and mobile phones has further enriched the HAR field. Multimodal approaches [43] offer a more comprehensive insights into human activities, particularly in scenarios where single modality sensing is inadequate.

Over the past decade, numerous reviews have focused on various aspects of HAR, such as human motion analysis, motion analysis in video data, posture prediction, HAR from 3D data, depth data, vision-based data, sensor-based HAR, wearable sensor-based HAR, human pose estimation, skeleton-based HAR, and RFID-based HAR [44,45,46,47,48,49,50,51,52,53]. Moreover, several reviews have reviewed the diverse datasets and modalities used in HAR [54,55,56,57]. Despite this extensive body of work, a comprehensive examination encompassing a wide array of data modalities including silhouettes, point clouds, range sensors, RGB, depth, skeleton, infrared sequences, event streams, motion capture devices (Mocap), acoustic sensors, wearable devices, radar, Wi-Fi, and sensor fusion remains limited. This review aims to fill this gap by providing an integrated perspective on the latest advancements in HAR across these diverse modalities.

This review presents a thorough review of the latest breakthroughs in machine learning technologies which are tailored for improving HAR. It examines the significance of various methodologies proposed in the literature together with their applicability. The primary objective is to offer a comprehensive summary that highlights recent advancements as well as their implications. By combining insights from various studies, this review provides a cohesive view of current HAR technologies. This review also identifies unresolved issues and limitations within existing approaches that underscore critical needs and urge for further research to develop solutions that bridge these gaps. By identifying potential areas for exploration, this review serves as a roadmap for advancing the field of human activity recognition and fostering a deeper understanding of both its current capabilities and future potentials.

This review paper is carefully structured into six sections to provide a systematic exploration of human activity recognition using machine learning and deep learning techniques. After this introduction, fundamental concepts and historical developments within HAR is presented in Section 2 which sets up the stage for deeper discussions. Various data modalities that are used for HAR research are reviewed in Section 3 which also details the specific sensors and technologies that are instrumental in advancing the field. Section 4 explores the publicly available datasets that are pivotal for training and evaluating HAR systems, along with a discussion on their characteristics and applications. Different machine learning techniques which are frequently applied to HAR are analysed in Section 5, which also highlights contributions and comparative evaluation. Subsequent sections identify the challenges and research gaps in the field of HAR and also propose potential research directions that could address these. This review concludes by presenting a summary of the findings and their implications for future technology development in human activity recognition, emphasizing the continued need for innovation and interdisciplinary collaboration.

## 2. Review Methodology

This section details the systematic review process undertaken for this study, describing the eligibility criteria for the study selection, search strategy, and screening process. This ensures transparency and reproducibility. The methodology follows the Preferred Reporting Items for Systematic Reviews and Meta-Analysis (PRISMA) 2020 [58,59] guidelines for systematic reviews.

### 2.1. Eligibility Criteria

#### 2.1.1. Inclusion Criteria

An article was considered for this review if any of the following conditions are satisfied:The article either investigates or proposes methods for HAR.It uses sensor data relevant to HAR, such as RGB, depth, infrared, motion capture devices, skeleton joint positions, wearable devices, acoustic sensors, radar, WiFi, LiDAR, proximity sensors, or combinations of these.It applies machine learning or deep learning techniques for activity classification.The article is published in English in a peer-reviewed journal or conference proceeding.

#### 2.1.2. Exclusion Criteria

An article was excluded from this review if it met any of the following conditions:The article focuses on sensor development without direct relevance to HAR.The article does not employ machine learning or deep learning for activity recognition.The article primarily focuses on human detection, pose estimation, or motion identification without activity classification.Reviews and non-peer-reviewed, non-English articles.

### 2.2. Search Criteria

To identify relevant studies, a comprehensive search was conducted across the following databases up to 30 August 2024:ScopusGoogle ScholarIEEE Xplore Digital LibraryACM Digital Library

The following keyword combinations were used to query each database during the search process:Activity-related keywords: [“human activity recognition” OR “activity monitoring” OR “gesture recognition” OR “Activity Discovery”].Machine learning-related keywords: [machine learning OR deep learning OR convolutional neural network OR recurrent neural network OR support vector machine OR random forest OR Reinforcement Learning OR Graph Convolutional Networks OR Generative Adversarial Networks OR Autoencoders OR Artificial neural network OR ANN].Sensor-related keywords: [RGB OR Depth OR Infrared OR Motion capture devices OR Mocap OR Skelton joint positions OR Wearable devices OR accelerometer OR gyroscope OR Acoustic sensors OR Radar OR Wifi OR Lidar OR Proximity sensors OR Fusion HAR].

### 2.3. Screening Strategy

The PRISMA 2020 guidelines were adopted for the article selection process, as illustrated in Figure 1 (PRISMA flowchart). The selection process consisted of four stages:Identification:The initial search yielded 7938 records from the seven databases: ACM (479), IEEE Xplore (4760), Elsevier (563), Springer (1570), Wiley (18), Taylor and Francis (32), and Nature (286).Screening:A total of 5759 records were screened based on titles and abstracts to remove studies that were not relevant to the scope of this review. In total, 1949 records were excluded for not utilising machine learning or deep learning techniques for HAR. Additionally, 3890 articles were found to be inaccessible due to paywall restrictions, despite utilizing institutional subscriptions and exploring open-access alternatives. These constraints limited access to potentially relevant studies, but efforts were made to ensure that the available open-access literature provided a comprehensive and representative dataset.Eligibility:Following the initial screening, 1869 full-text articles were assessed for eligibility based on the predefined inclusion and exclusion criteria. During this stage, 650 articles were excluded for reasons such as lack of implementation details, a narrow focus on partial-body movements, or the absence of performance metrics essential for evaluating HAR systems. Additionally, 860 articles were excluded due to significant overlap with existing studies, ensuring that only unique and methodologically sound research was retained for review.Inclusion:After completing the eligibility assessment, 359 studies met the inclusion criteria and were included for detailed analysis. These studies represent a broad range of sensor modalities, machine learning methodologies, and emerging trends in HAR research. The selected studies provide valuable insights into the field, addressing key challenges and identifying potential research directions.

## 3. Human Activity Recognition (HAR)

### 3.1. Definition and Categories of Human Activity Recognition

#### 3.1.1. Overview of Human Activity Recognition

Human Activity Recognition (HAR) is the automated process that can identify and interpret human actions using a sensing modality and a learning algorithm. HAR involves several stages including data capturing, data processing, and analysing data from various types of sources. Sensors which are utilised to classify and understand human actions often include devices such as wearable sensors, cameras, and depth sensors [55]. The primary objective of any HAR system is to interpret a diverse range of human activities from gestures to complex sequences like dancing to improve our understanding of human activities in different contexts [60].

Leveraging advanced sensors and computational techniques, HAR enables the development of sophisticated systems capable of perceiving, predicting, and responding to human activities. These capabilities are pivotal for driving progress in several domains, including healthcare [61], smart environments [62], robotics [63], surveillance, and human–computer interaction [64]. The interdisciplinary nature of HAR integrates elements from computer vision, signal processing, machine learning, and sensor technologies, enabling the accurate recognition and interpretation of human activities.

HAR significantly impacts a wide range of sectors, including healthcare, security and surveillance, smart homes, autonomous driving, entertainment, sports analytics, industrial automation, and education. The diverse applications of HAR in these fields are illustrated in Figure 2, showcasing the transformative role of this technology in enhancing our understanding and interaction with human activities.

#### 3.1.2. Categories of Activities

Human activities encompass a broad spectrum of actions and behaviours that can be classified into several categories based on their nature and complexity, as illustrated in Figure 3. These categories, widely recognised in the literature [65,66], provide a structured framework for understanding the various types of activities performed by humans.

Gestures represent the most fundamental form of human activity, involving intentional body movements or signals made with the hands, arms, or entire body to convey information, commands, or expressions. These can range from simple gestures, such as waving or pointing, to more intricate movements used in sign language or other communication interfaces [67,68].

Building upon these basic gestures are atomic actions, which are simple, fundamental movements that are performed independently. Examples include walking, running, sitting, standing, jumping, or crouching. These atomic actions often serve as the foundational elements for recognising more complex activities and behaviours [69].

Interactions elevate the complexity of human activities by involving engagements between humans or between humans and objects. Characterised by a dynamic exchange of information, actions, or responses [70,71], human-to-human interactions include communication, cooperation, or collaboration, such as a conversation between two people. Human–object interactions [65,72], on the other hand, involve individuals engaging with physical objects or systems, such as manipulating tools or interacting with technology.

When multiple individuals coordinate their actions, they participate in group activities. These activities often require synchronisation, cooperation, and communication among participants to achieve shared goals or outcomes [73,74]. Examples of group activities include team sports like football or volleyball [75], performing in an orchestra, participating in collaborative projects, or engaging in collective performances [76,77].

Behaviours encompass a broad range of actions, expressions, and responses exhibited by individuals or groups. This category includes social behaviours, emotional expressions, habitual actions, decision-making patterns, and psychological responses observed in various contexts.

Events are more complex forms of human activities that typically involve multiple individuals and occur within specific environments. Examples of such events include music festivals, concerts, and weddings, where various activities take place simultaneously and involve numerous participants.

Based on the nature of motion or postures, activities can be further categorised into static actions, transitional actions, and dynamic actions. Static actions involve minimal movement, such as sitting and reading a book, whereas transitional actions involve a change in posture, such as sitting down or standing up. Dynamic actions are more complex and involve significant motion, such as walking and running. Activities can also be grouped into broader categories, including daily living activities, remote health monitoring tasks, sports actions, and activities related to security and safety, such as suspicious or dangerous activities.

### 3.2. Stages of HAR

The HAR process typically starts with data acquisition, where raw sensor data are collected from various sources, such as accelerometers, gyroscopes, or cameras, to capture human movements and actions. This stage involves selecting the appropriate sensors based on the type of activities and the environment in which they occur, determining suitable sampling rates, and ensuring data quality to facilitate accurate activity recognition. Figure 4 illustrates the various stages involved in the HAR process.

Following data acquisition, the next stage is data pre-processing, which aims to improve data quality and extract the relevant features necessary for activity classification. Pre-processing techniques include data cleaning to remove irrelevant or erroneous data, data filtering to eliminate noise, normalisation to standardise the data, and background removal in the case of video data. Additional techniques such as feature extraction, data segmentation, feature selection, and dimensionality reduction are applied to prepare the data for model training and to enhance the accuracy of subsequent activity recognition stages.

Once the data have been pre-processed, the focus shifts to model selection and training. In this stage, the most suitable machine learning or deep learning algorithms are chosen based on the nature of the data and the specific activity recognition tasks. This involves configuring model architectures, optimising hyperparameters, and training the model using labelled training data to develop robust models capable of accurately classifying activities from sensor data in real time. In cases where labelled data are unavailable, unsupervised methods are applied to estimate the number of activities in the dataset. Techniques such as clustering are used to group similar data points, after which class labels are assigned by examining cluster centres or random cluster members [76].

Following model training, fine-tuning techniques such as hyperparameter optimisation and transfer learning are employed to further refine model performance. Hyperparameter tuning involves adjusting parameters like learning rate and batch size to improve model accuracy, while transfer learning leverages pre-trained models to enhance performance on new but related tasks, reducing the need for large amounts of labelled data.

The final stage is activity recognition, where the trained model is deployed to classify real-time sensor data and infer human activities. Once activities are accurately identified, the information can be used in various applications to perform autonomous actions, such as alerting emergency response teams or notifying a family member in case of a fall. This ability to interpret and act upon human activities in real time highlights the transformative potential of HAR across numerous domains.

### 3.3. Activity Discovery and Activity Recognition

Various pervasive applications are made possible by human activity recognition, which converts noisy, low-level sensor input into high-level human activities [77]. The understanding of human activities includes both activity recognition and activity pattern discovery [78]. Activity recognition is centred on the precise recognition of human activities using a predefined activity model. Thus, a human activity recognition researcher first develops a high-level conceptual model, which is subsequently put into practice by creating an appropriate pervasive system [78]. Two assumptions are made in an activity recognition process: firstly, the set of activities is fixed, and the user activity model is fixed. However, both fall short when a system is deployed in an open and flexible settings, for an extended period of time, and target a broader spectrum of user. First of all, it is impractical to presume that only the pre-defined set of activities will be performed by users over an extended period of time. The pre-trained model may misclassify or fail to recognise a novel activity performed by a user if it was not observed or learned during the training period. Secondly, as user activities change over time, it can be challenging, or even infeasible, to collect all possible user activities during the training period. Both of the aforementioned scenarios may cause activity detection algorithms to perform worse, which could provide users with unwanted services.

Conversely, activity pattern discovery focuses more on immediately identifying unknown patterns from sensor data without the need for pre-established models or presumptions. In order to find activity patterns, a researcher that specialises in activity pattern detection first creates a ubiquitous system and then examines the sensor data. Despite their differences, the two approaches share the goal of advancing human activity technology [78]. They also work well together since the identified activity pattern can be used to define subsequent activities that can be identified and monitored.

The steps involving human activity recognition and activity pattern discovery are depicted in Figure 5 and Figure 6, respectively. Both the activity recognition and activity discovery process start with segmented activities; feature extraction/selection is an intermediate step which is followed by the activity representation step where features from consecutive frames are combined. Activity recognition requires the selection of an appropriate learning algorithm while activity discovery uses a clustering algorithm to estimate the number of activities in the dataset which are then annotated for classification. Methods often used by human activity discovery researchers are tabulated in Table 1.

### 3.4. Types of Sensors Used for Human Activity Recognition

Human activity recognition (HAR) methods can be broadly categorised into several categories based on the types of sensors used: non-visual sensors, visual-based sensors, context-aware systems, and hybrid approaches [69]. Each category leverages different modalities to capture and analyse human activities, offering unique advantages and addressing specific challenges associated with activity recognition. Understanding the strengths and limitations of these sensor types is crucial for developing effective HAR systems tailored to diverse applications and environments. Figure 7 provides an overview of the various HAR categories and their corresponding sensor modalities.

Different modalities of HAR, each with their respective advantages and limitations, are summarised in Table 2. By combining multiple modalities, hybrid methods aim to mitigate the drawbacks of single sensor approaches and improve the robustness and accuracy of activity recognition systems. Hybrid methods are typically characterised into three categories such as early fusion, late fusion, and hybrid fusion. In early fusion raw data or low-level features are combined at an early stage for HAR. While in the late fusion, each sensor modality is processed independently, and classification results are combined final classification. Hybrid fusion combines elements from both early and late fusion for improved classification.

## 4. Data Collection and Pre-Processing

In human activity recognition (HAR), the accuracy and reliability of the recognition systems heavily depend on the quality and diversity of the data used for model training and testing. Data collection plays a critical role in capturing relevant information from different sources, while data pre-processing ensures that this information is clean, structured, and ready for analysis. This section delves into both the publicly available datasets for HAR, categorised by different sensing modalities, and the various pre-processing techniques used to optimise the data for machine learning models.

### 4.1. Publicly Available Datasets for Human Activity Recognition

Publicly available datasets are essential resources for advancing and benchmarking research in human activity recognition (HAR). These datasets span a broad spectrum of data modalities, including vision-based, sensor-based, skeleton-based, and multimodal datasets.

Vision-based datasets typically consist of RGB images or video streams that capture human movements. RGB datasets allow researchers to analyse visual patterns and behaviour with the surrounding context. On the other hand, depth-based datasets include 3D spatial information from sensors such as Kinect or LiDAR. For instance, 3D datasets offer detailed data about object and human positioning in space. Sensor-based datasets are collected from sensors such as the accelerometers, gyroscopes, or inbuilt smartphone sensors. Sensor-based data provide motion and orientation data that are critical for activity recognition. Skeleton-based datasets focus on tracking key joint positions of the human body which simplify the representation of human motion while still enabling accurate activity analysis. Multimodal datasets integrate various data streams such as RGB, depth, infrared, and skeleton coordinates. Multimodal datasets provide a comprehensive and enriched dataset for recognising complex human activities and it is for scenarios where one sensing modality may seem inadequate. Key publicly available datasets for HAR from each category are listed in Table 3. The datasets that are mostly used from each sensor type with most subjects’ action classes are listed here; for a comprehensive list of publicly available HAR datasets, please refer to Appendix A.

### 4.2. Data Pre-Processing

Data pre-processing is a critical stage in human activity recognition (HAR) that significantly enhances the quality, consistency, and usability of sensor data [124]. As raw data collected from various sensors can be noisy, incomplete, or inconsistent, effective pre-processing is essential for preparing the data for subsequent analysis and model development. Pre-processing techniques include a series of steps aimed at cleaning, transforming, and organizing data, ensuring it is suitable for activity recognition tasks [83].

#### 4.2.1. Importance of Data Pre-Processing in Human Activity Recognition

Regardless of the sensing medium, data pre-processing plays a pivotal role in ensuring the quality and reliability of human activity recognition (HAR) systems [124]. It acts as a critical foundation for enhancing the accuracy and performance of models. It is essential for cleaning, organizing, and transforming raw sensor data into a usable format [124].

Similarly, data pre-processing is essential in RGB-based HAR, where raw video frames captured by RGB cameras often contain a wide range of unnecessary information, such as background clutter, lighting variations, and irrelevant movements [116]. Pre-processing techniques, such as background subtraction, frame cropping, and normalisation, help isolate human subjects, remove extraneous visual noise, and standardise colour channels for uniform input [125].

#### 4.2.2. Data Pre-Processing Techniques

Data pre-processing involves a variety of techniques that are designed to transform raw and unprocessed data into a structured and refined format. Processed data ultimately enhance the accuracy and performance of human activity recognition (HAR) systems [126].

##### Data Preparation

Data preparation is a foundational step in pre-processing that involves two key tasks: activity segmentation and noise removal.

Activity segmentation is the process of dividing continuous data streams into smaller, manageable sections that correspond to specific actions or activities. This segmentation allows researchers to isolate and analyse individual activities more effectively, leading to more accurate labelling and classification of human actions [127]. Figure 8 illustrates an example where continuous human activities (top row) and hand joint movements over time (bottom row) are depicted.

Noise removal is equally important for HAR, as it involves eliminating unwanted disturbances from the raw data. If left unprocessed, these artifacts can obscure meaningful patterns and introduce inaccuracies during analysis. Noise removal techniques often include filtering and smoothing [128].

##### Feature Engineering

Feature extraction involves identifying relevant patterns from raw data that are indicative of specific human activities [129]. In human activity recognition (HAR), extracted features may include statistical measures such as mean, variance, or skewness [129], frequency domain characteristics like spectral entropy [62], or temporal patterns such as joint angles [129], joint positions [130], and joint orientations [131].

Once the relevant features have been identified, feature selection techniques are applied to retain only the most useful ones, while discarding irrelevant, redundant, or noisy features [132]. This step is crucial for reducing the dimensionality of the data and simplifying the subsequent analysis. Common methods for feature selection include techniques such as Principal Component Analysis (PCA) [83], Recursive Feature Elimination (RFE) [133], Gaussian filters [134], and various filtering approaches [135] that rank features based on their importance to the task.

##### Data Transformation and Standardisation

Data transformation and standardisation ensure consistency and comparability across different features or data types [136]. These techniques help stabilise training processes, reduce biases caused by varying data scales, and improve the performance of human activity recognition (HAR). Scaling and normalisation involve adjusting the range of input features to a standard scale, such as normalising pixel values between 0 and 1, or standardising features to have a mean of 0 and a standard deviation of 1 [137].

##### Data Enhancement and Handling Class Imbalances

Data enhancement and addressing class imbalances are critical for improving the performance of deep learning-based models in human activity recognition (HAR) [138]. Deep learning models require large amounts of diverse labelled data to generalise well [139]. Data augmentation is a key technique used to artificially expand the dataset by generating additional training samples through transformations [140] such as changing illumination, adding noise, applying rotations, translations, flips, crops, scaling, and zooms to the original data [141]. These transformations help diversify the training data, allowing models to learn from different variations of the same activity and thus reducing overfitting.

For sequential data, such as time-series data from sensors or video sequences, techniques like padding and trimming ensure that all sequences have uniform feature dimensions [142], which is necessary for efficient batch processing in deep learning models. Padding involves appending zeros to shorter sequences, while trimming longer sequences to a fixed length.

Another crucial component of data improvement is the encoding of categorical variables which involves converting categorical variables into numerical representations using methods like one-hot encoding. Encoding allows deep learning models to process non-numerical data effectively [143]. This is particularly useful in HAR when dealing with labels or other categorical variables that need to be converted into a format compatible with machine learning algorithms.

A common challenge in HAR datasets is class imbalance where certain activities occur more frequently than others. Class imbalance often leads to biased models. To address this, HAR researchers often use techniques like oversampling minority classes or under-sampling majority classes [143]. Oversampling is helpful to generate additional data points for underrepresented activities. Under-sampling reduces the data points from overrepresented activities. These approaches help achieve a balanced dataset which is crucial for ensuring that the model does not favour the majority class, and the model can effectively learn to recognise minority class activities.

## 5. Machine Learning Techniques in Human Activity Recognition

Before the widescale adoption of machine learning techniques, traditional approaches to human activity recognition (HAR) relied heavily on rule-based systems [144]. These methods were designed to detect and classify activities based on predefined criteria and certain threshold values. For example, to identify accelerometer-based activities, thresholding techniques were employed to recognise specific motions like walking and running by detecting changes in acceleration patterns that crossed predetermined limits [145]. Similarly, in time-series analysis, methods like peak detection or pattern matching were also used to identify repetitive activities such as cycling or repetitive arm movements.

The sections below provide a detailed overview of the various machine learning techniques employed in HAR, their methodologies, applications, and performance benefits, emphasising the role of both shallow learning and deep learning models in advancing the field.

### 5.1. Shallow Learning Models

#### 5.1.1. Supervised Learning Techniques

Supervised machine learning techniques are fundamental to human activity recognition (HAR) as they rely on labelled datasets to train models for classifying human activities from various sensor modalities. These techniques are highly effective in extracting and learning patterns from data sources such as RGB images, wearable devices, mobile phone sensors, depth cameras, and skeletal tracking systems. By utilizing features extracted from these modalities, supervised learning algorithms can discern and classify distinct human actions with high accuracy.

Popular supervised learning algorithms in HAR include Decision Trees (DT), Random Forests (RF), Support Vector Machines (SVM), K-Nearest Neighbours (KNN), and gradient boosting methods like XGBoost and NGBoost. Each algorithm has its strengths and trade-offs depending on the sensor data and application scenarios.

Decision Trees (DT) [146,147,148,149,150] are widely used for classifying activities by partitioning the feature space based on threshold values. They are effective in handling continuous and categorical sensor data, such as accelerometers and gyroscope readings, as well as frame-wise features extracted from RGB images. However, a key limitation of Decision Trees is their tendency to overfit, especially when the tree becomes too deep, leading to poor generalisation on complex data.

Random Forest (RF), an ensemble method that combines multiple decision trees, help to mitigate overfitting and improve the overall accuracy of HAR systems [151,152,153,154,155,156]. RF models excel in capturing more complex patterns from high-dimensional sensor data and are commonly used in applications requiring robust classification. However, RF models are computationally expensive, especially when the number of trees is large, and they can suffer from interpretability challenges, particularly in the presence of highly imbalanced datasets.

Support Vector Machines (SVMs) are well-suited for HAR involving complex boundary separations in high-dimensional feature spaces [157,158,159,160,161,162]. They effectively utilize kernel methods to manage non-linear relationships between features from different sensors, making them ideal for differentiating activities like walking and running. However, SVMs are sensitive to kernel and hyperparameter configurations, and tuning them for optimal performance can be computationally intensive, particularly for large datasets.

K-Nearest Neighbour (KNN) is a simple yet powerful instance-based learning algorithm. It classifies activities based on the majority vote of the nearest data points in the feature space, which makes it intuitive and effective for HAR applications. However, KNN becomes computationally expensive for large datasets and struggles with performance when distinguishing between complex or overlapping activity patterns [163,164,165,166].

Hidden Markov Models (HMMs) are utilized to model sequential dependencies in HAR data, making them particularly useful for recognising activities like walking, running, or sitting, where temporal patterns are crucial. HMMs efficiently handle state transitions in time-series data, but their assumptions of fixed states and transition probabilities limit their flexibility in capturing continuous or highly variable human activities [167,168,169].

Naive Bayes classifiers are used for efficient classification in HAR by assuming conditional independence among features. They perform well in scenarios where this assumption holds true, but in real-world applications, sensor features are often correlated, which can reduce the model’s accuracy. Naive Bayes classifiers are commonly applied to sensor data from accelerometers, gyroscopes, magnetometers, and skeletal tracking systems [170,171,172,173].

Gradient boosting methods, such as XGBoost and NGBoost [174,175,176], have gained traction in HAR research for their ability to handle complex, multi-sensor data with high accuracy. These algorithms iteratively refine predictions to minimize errors, making them particularly effective for imbalanced datasets. However, gradient boosting methods are sensitive to hyperparameter tuning and require substantial computational resources, especially for large datasets.

Stochastic Gradient Descent (SGD) is commonly used in HAR for optimizing linear models, neural networks, and SVMs [177,178,179]. It updates model parameters incrementally based on small batches of sensor data, making it efficient for large-scale HAR applications. However, careful tuning of the learning rate and other hyperparameters is necessary to avoid convergence problems.

Supervised learning models are typically trained using labelled data, where a portion of the dataset is used for training, while the remaining data are reserved for testing and validation [41]. The models improve their ability to generalise across diverse activities using an iterative optimisation process. Hyperparameter tuning and cross-validation are frequently applied during the process. The main advantage of supervised learning in HAR is its ability to learn complex mappings between input features and activity labels. These methods lead to robust and real-time recognition performance across various environments. Despite their effectiveness, supervised learning methods require a substantial amount of labelled data for training [180]. The performance can degrade where labelled data are scarce; also, factors like high variability in activity contexts leads to poor performance [181].

Nonetheless, they continue to be widely used in HAR systems powering applications ranging from healthcare monitoring, rehabilitation, and human–computer interaction to smart environments. The versatility of supervised learning algorithms allows them to integrate information from multiple sensor modalities, enhancing the robustness and accuracy of activity recognition systems. Appendix A provides an overview of various machine learning-based HAR techniques across different sensor modalities, showcasing the versatility of supervised learning methods in this domain.

#### 5.1.2. Unsupervised Learning and Clustering Techniques

Unsupervised machine learning methods offer a valuable approach to human activity recognition (HAR) by uncovering hidden patterns and structures within sensor data without the need for labels [182,183]. These methods are especially useful when labelled data are scarce or unavailable, allowing the HAR system to autonomously learn activity patterns directly from raw data. In HAR systems utilising modalities such as RGB video, mobile phones, wearable sensors, depth sensors, Mocap, and skeleton data, unsupervised learning algorithms like clustering or dimensionality reduction techniques—such as Principal Component Analysis (PCA) and t-distributed Stochastic Neighbour Embedding (t-SNE) [83]—are commonly employed to identify underlying patterns within the data.

Clustering techniques, including K-means [184,185,186,187], Spectral Clustering [188,189], Agglomerative Clustering [188], DBSCAN [190], and Gaussian Mixture Models (GMM) [191,192], enable the grouping of similar activities into clusters by detecting natural groupings based on the similarity of features. These methods are essential for revealing inherent similarities and differences among various activities and sensor modalities, helping to cluster activities that share similar characteristics. For example, clustering algorithms can automatically group activities like walking, running, and sitting by analysing motion data collected from accelerometers and gyroscopes in wearable sensors.

Unsupervised learning techniques are particularly beneficial for anomaly detection and activity discovery in HAR. These methods allow the system to detect unknown or abnormal activities without requiring labelled examples. For instance, in healthcare settings, unsupervised learning can identify anomalous patterns in patient behaviour, such as unexpected falls or irregular walking patterns, which may signal health issues. Similarly, deep clustering [188], particle swarm optimisation, and game theory [83] methods have been explored to discover new activity patterns in HAR datasets, offering the potential for automatic identification of previously unseen or unlabelled activities.

By leveraging complementary information from various sensor modalities, unsupervised learning enables HAR systems to identify, group, and characterise activities based on their inherent similarities, contributing to applications such as anomaly detection, behaviour profiling, and activity summarisation. The flexibility of unsupervised learning techniques makes them invaluable for cases where labelled data are limited or unavailable, and they help the HAR system to generalise better across different activity contexts and sensor environments. Appendix A summarises various unsupervised learning techniques and their applications in HAR using different modalities.

#### 5.1.3. Semi-Supervised Learning Techniques

Semi-supervised learning methods strike a balance between supervised and unsupervised approaches, offering a pragmatic solution for human activity recognition (HAR) in situations where labelled data are limited or expensive to obtain [193]. These techniques leverage both labelled and unlabelled data, which are particularly advantageous in real-world applications where acquiring extensive labelled datasets is often challenging, but abundant unlabelled data are readily available. This hybrid learning paradigm is well-suited for HAR systems that utilise multiple data modalities, such as RGB video, mobile phones, wearable sensors, depth sensors, and skeletal data in smart home environments [194].

One innovative application of semi-supervised learning in HAR is the Federated Learning (FL) framework proposed by Yang et al. [193]. Their approach, FedHAR, addresses data scarcity by combining federated, semi-supervised, and transfer learning techniques. In this framework, a global model is trained using a small amount of labelled data from each user and subsequently personalised for individual users through transfer learning. FedHAR was validated on the MobiAct and WISDM datasets, demonstrating its effectiveness in adapting to personalised activity recognition scenarios.

A semi-supervised adversarial learning approach using Long Short-Term Memory (LSTM) networks for human activity recognition (HAR) is proposed by the authors in reference [194]. This method used both labelled and unlabelled data to train the model, enhancing its ability to adapt to new activities and changes in activity routines. The evaluation was conducted on datasets collected from smart home environments with heterogeneous sensors, specifically the Kasteren and CASAS datasets [195]. The proposed method achieved an accuracy of over 95% which surpassed existing state-of-the-art methods.

A semi-supervised active transfer learning (SATL) method for human activity recognition (HAR) to reduce the burden of manual data labelling is proposed by Oh et al. [196]. The method used previously learned data to reduce labelling tasks on new data. A DNN-based basic model was trained on an initial training set and used to construct a correct classifier model. Both models are then used to analyse unlabelled data and select the most informative samples for labelling by a human annotator. The method was evaluated on the HCI-HAR and mHealth dataset, achieving over 95% accuracy while significantly reducing the amount of data requiring manual labelling. A context-aware semi-supervised method was proposed by [197] which used the HCI-HAR [198], WISDM, PAMAP2 [199], mHealth [200] datasets for model validation.

Techniques like self-training, co-training, and graph-based semi-supervised learning iteratively refine model predictions by incorporating information from both labelled and unlabelled samples [201]. In self-training, a model is initially trained on a small amount of labelled data, and the confident predictions on unlabelled data are iteratively added to the training set [201]. Co-training involves training multiple models on different views or subsets of the data, where each model helps label the unlabelled data for the other. Graph-based methods use graph representations to propagate label information from labelled to unlabelled data based on their similarity. A semi-supervised method for HAR using multimodal data from Instagram, particularly image and text data, is proposed by Kim et al. [201]. The goal was to improve recognition accuracy while using a limited amount of labelled data, leveraging the abundance of unlabelled data on social media. The proposed method achieved a recognition accuracy of 71.58%, outperforming existing HAR methods on multimodal datasets.

Semi-supervised ensemble learning for HAR using the CASAS Kyoto dataset was explored by Patricia et al. [202]. The researchers applied a distance-based clustering analysis method to identify clusters of activity characteristics, which then served as the input for a supervised classification process. Evaluation using quality metrics indicates favourable results, with the average accuracy exceeding 95% when using the Agglomerative Clustering algorithm with the Bagging classifier. This paper [203] focused on leveraging unlabelled data to improve the performance of Convolutional Neural Networks (CNNs) in HAR. Two semi-supervised architectures, CNN-Encoder-Decoder and CNN-Ladder, were used for this purpose. The study evaluated the proposed method on three datasets: ActiTracker, PAMAP2, and mHealth. The paper reported up to an 18% improvement in F1-score compared to previous baseline methods.

A semi-supervised framework that used generative adversarial networks (GANs) with temporal convolutions for human activity recognition (HAR) is suggested by Hazar et al. [204]. The framework aimed to overcome the restrictions faced by conventional approaches, particularly the reliance on large, labelled datasets. The framework was evaluated using the PAMAP2, Opportunity-locomotion, and LISSI HAR datasets under real-world semi-supervised scenarios, examining factors such as inter-subject training, labelled data amount, class number, and IMU positioning effects on performance. Results demonstrated high classification performance and generalisation ability, with 25% improvement compared to baseline methods when limited annotated data were provided.

By combining the strength of labelled data with the abundance of unlabelled data, semi-supervised methods help the model generalise more effectively across diverse activity contexts and sensor modalities [197]. This approach enhances the robustness and scalability of HAR systems, enabling them to adapt to new activities, sensor configurations, and unseen environments with minimal supervision. Semi-supervised learning also allows for more accurate and reliable activity recognition in real-world applications like healthcare monitoring, smart environments, and human–computer interaction, where labelled data are often limited [194]. Various techniques in semi-supervised HAR are tabulated in Appendix A showcasing their applications across multiple modalities.

### 5.2. Deep Learning Models

#### 5.2.1. CNN, RNN, GCN, and LSTM-Based Methods

Deep learning-based methods have become the cornerstone of human activity recognition (HAR) across various sensing modalities, such as RGB [205], mobile phones [206], wearable sensors [207], depth sensors [208], and skeleton data [209]. This shift from traditional machine learning to deep learning approaches started gaining significant momentum in the mid-2010s when researchers realised the potential of deep neural networks to automatically learn hierarchical representations from raw sensor data without manual feature engineering [210]. Key deep learning architectures include Convolutional Neural Networks (CNNs) [15], Recurrent Neural Networks (RNNs) [19], Graph Convolutional Networks (GCNs) [21], Long Short-Term Memory (LSTM) [211], and Transformers architectures. Each architecture serves a distinct purpose: CNNs are effective for spatial data analysis, such as images or depth maps, while RNNs, LSTMs, and GCNs excel at capturing temporal dynamics or sequential dependencies in time-series data, such as sensor readings from wearables, mobile phones, or skeleton-based data. To capture long-range dependencies and the global context in sequential data, transformers are used for HAR due to their ability which make them effective for modalities like skeleton sequences, images, and time-series sensor data. The ability of these networks to process raw input directly, bypassing manual feature extraction, allows for more efficient recognition of complex human activities across diverse environments and modalities.

Deep learning models in HAR have greatly enhanced the field’s ability to interpret multimodal sensor data, enabling the extraction of more meaningful and robust features [62]. For example, CNNs were applied to process spatial features from images by authors in reference [212]. The authors in [72] applied RNNs to capture temporal relationships in time-series data. These models can be integrated into multimodal frameworks that fuse information from different sensors. Visual data from RGB cameras combined with accelerometer or gyroscope readings [213] can be used by these hybrid methods. This multimodal approach enhances accuracy and robustness, allowing HAR systems to perform more effectively in real-world applications, including healthcare monitoring [61], smart environments [214], and human–computer interaction [215]. A key advantage of deep learning over traditional methods is its ability to automatically learn complex, hierarchical features from large datasets, which are often necessary to capture the subtleties and intricacies of human activities [216]. This is particularly valuable in HAR scenarios where large-scale, diverse, and complex data are involved. Deep learning methods can generalise well across varied contexts and adapt to new activities or environments, making them more versatile and reliable.

In recent years, several advancements in deep learning have been introduced to further enhance HAR capabilities. The authors in reference [205] introduced a method for 3D human pose estimation from a single image using a Convolutional Neural Network (CNN) and transfer learning. To improve the accuracy of 3D pose estimation, the authors leveraged a large-scale 2D pose dataset. The methodology involves training a CNN to predict the 2D pose and then lifting the 2D predictions to 3D using a separate network. This study [94] introduced a novel dataset called DHP19 for 3D human pose estimation using dynamic vision sensors (DVSs). The dataset consists of synchronised recordings from four DVS cameras and a Vicon motion capture system. The paper presented a reference study on 3D human pose estimation using the DHP19 dataset and reported a 3D mean per joint position error or MPJPE of 87.54mm on the test set, which significantly reduced joint estimating errors with a reduced frame rate.

Convolutional Neural Networks (CNNs) have been used for human action recognition from 3D joints. For instance, a scene flow to action map description for RGB+D-based action recognition with CNNs was proposed by Wang et al. [217]. Here, 3D skeleton data were transformed to ten feature arrays and used as the input to CNNs for action recognition by Ke et al. [218]. In order to transform the visual appearance of human body parts obtained from various perspectives to a view-invariant space for depth-based human object interaction analysis, Rahmani et al. [219] created a deep CNN model.

A novel method called Spatial Temporal Graph Convolutional Networks (ST-GCNs) for skeleton-based action recognition was presented in this study [220]. The model works by constructing a set of spatial-temporal graph convolutions on the skeleton sequences to capture motion information. The authors evaluated the ST-GCN method on the Kinetics and NTU-RGB+D datasets. On Kinetics, ST-GCN achieved a top 1% accuracy of 30.7% and a top 5% accuracy of 52.8%, while on NTU-RGB+D, the model outperformed previous state-of-the-art skeleton-based models with cross-subject accuracy of 81.5% and cross-view accuracy of 88.3%. A new video-level framework called Temporal Segment Network (TSN) for action recognition in videos is proposed by [221] which aimed to model long-range temporal structures with a segment-based sampling and aggregation module. The model was evaluated on five challenging video-based action recognition benchmarks: HMDB51, UCF101, THUMOS14, ActivityNet v1.2, and Kinetics400. TSN achieved a state-of-the-art performance on these datasets, with accuracies of 71.0%, 94.9%, 80.1%, 89.6%, and 75.7%, respectively.

Actional-Structural Graph Convolutional Networks (AS-GCN) for skeleton-based action recognition was introduced in this work [222]. AS-GCN incorporated the actional links between joints in consecutive frames, in addition to the structural links between joints in a single frame. The model was evaluated on the NTU-RGB+D dataset and achieved an accuracy of 86.8% and 94.2% on the cross-subject and cross-view benchmark.

Richly Activated Graph Convolutional Network (RA-GCN) for robust skeleton-based action recognition was suggested by authors in this study [223]. RA-GCN applied a multi-stream framework with an activation module to handle noisy or incomplete skeleton data. The model was evaluated on the NTU RGB+D 60 and 120 datasets and outperforms state-of-the-art methods, particularly in scenarios with occlusion and jittering with an accuracy score of 87.3% and 81.1% on cross-subject settings.

Another graph-based method called Two-Stream Adaptive Graph Convolutional Networks (2S-AGCNs) for skeleton-based action recognition was proposed in this study [224]. The 2S-AGCN method used both joint and bone information by constructing two adaptive graphs, one for each type of information. The model was evaluated on the Kinect300 and NTU RGB+D dataset. The method achieves state-of-the-art performance on both the cross-subject and cross-view benchmarks with an accuracy score of 88.5% 95.1%. On the Kinetics dataset, the top 1% accuracy was 36.1% and top 5% accuracy was 58.7%.

Skeleton-based action recognition using Directed Graph Neural Networks (DGNs) was introduced in this study [225]. DGNs explicitly modelled the dependencies between joints by considering the direction of information flow. The model was evaluated on the NTU-RGB+D and Skeleton-Kinetics datasets and achieved competitive performance compared to other state-of-the-art methods. Feedback Graph Convolutional Network (FGCN) is another method for skeleton-based action recognition [226]. FGCN proposed a feedback mechanism into graph convolutional networks, enabling low-level layers to access semantic information from higher-level layers. The model is evaluated on multiple datasets including the NTU-RGB+D60, NTU-RGB+D120, Northwestern-UCLA datasets and achieves state-of-the-art performance. Shift Graph Convolutional Network (Shift-GCN) for skeleton-based action recognition is proposed by the authors [227]. Shift-GCN was designed to capture the temporal evolution of the skeleton sequence by using temporal shift modules. The model was assessed on NTU RGB+D, NTU RGB+D -120, and Northwestern-UCLA datasets and achieved accuracy of 96.5%, 85.9%, and 94.6% respectively.

Decoupled Spatial-Temporal Attention Network (DSTA-Net) was introduced by Shi et al. [228] for skeleton-based action and gesture recognition. In the DSTA-Net, spatial and temporal features are decoupled using attention modules without the need for hand-crafted features. Additionally, the method also proposed using four data streams such as spatial-temporal, spatial, slow-temporal, and fast-temporal streams. DSTA-Net was evaluated on two hand gesture recognition datasets, SHREC [229] and DHG [230], and two human action recognition datasets, NTU-60 and NTU-120. On the hand gesture recognition datasets, the accuracies were 91.5%/96.4% while the accuracies on the action recognition datasets were 86.6% and 89.0%, respectively. Liu et al. [231] proposed a new method for skeleton-based action recognition that disentangles and unifies graph convolutions which consists of a disentangled multi-scale aggregation scheme to remove redundant dependencies between features in different neighbourhoods, and a unified spatial-temporal graph convolution operator (G3D) to facilitate direct information propagation between joints in both spatial and temporal domains. The method was assessed on the NTU RGB+D 120, NTU RGB+D 60, and Kinetics Skeleton 400 datasets.

Kwon et al. [232] explored a new perspective on self-similarity for learning generalised motion representation by proposing a neural block called SELFY, based on spatial-temporal self-similarity (STSS). The authors validated the effectiveness of SELFY on four RGB benchmark datasets: Diving-48, Something-Something-V1 and V2, and FineGym datasets with an accuracy of 41.6%, 84.4%, 91.1%, 87.7%, respectively. Davoodikakhki et al. [233] introduced human action classification using hierarchical classification, network pruning, and skeleton-based pre-processing to improve model robustness and performance. The method was tested on the NTU RGB 60, NTU RGB 120, N-UCLA, and UTD Action Dataset. Video-Pose Network (VPN++) for action recognition is proposed by Das et al. [234] that combines RGB video and 3D pose data through knowledge distillation. VPN++ uses two levels of distillation: feature-level and attention-level, to improve overall HAR performance. The authors evaluated the method on multiple datasets, including Smarthome dataset [235], NTU60, NTU120, and N-UCLA datasets with an average accuracy of 71.0%, 90.7%, 86.2%, and 93.5%, respectively.

Video Swin Transformer, a new type of vision transformer architecture for video recognition that uses shifted windows to compute self-attention across video frames, allowing efficient and effective modelling of long-range temporal information, is introduced by Liu et al. [236]. Experiments on several video recognition benchmarks, including the Kinetics-400, Kinetics-600, and Something-Something V2 with an average accuracy score of 84.9%, 85.9%, and 69.6% demonstrate improved performance compared to previous convolutional and transformer-based approaches. A framework for video recognition using webly-supervised learning from multiple data sources named OmniSource was introduced by Duan et al. [237] which uses a task-specific data collection strategy and a unified framework to leverage various web data formats and perform activity recognition.

Attention mechanisms [25], for example, have allowed models to focus on the most relevant parts of the input data, leading to more accurate and efficient activity recognition. Graph-based networks, such as GCNs [21], have shown promising results in capturing spatial and relational dependencies in skeleton-based activity recognition. Additionally, transfer learning has emerged as a key technique [238], enabling models trained on large, pre-existing datasets to be fine-tuned for specific HAR tasks with limited data. As deep learning continues to evolve, novel architectures, such as attention-based transformers and reinforcement learning frameworks, are being explored to improve HAR’s performance further. These methods have shown potential in overcoming challenges such as data imbalance, noise, and complex activity patterns, making HAR systems more scalable and adaptable to real-world environments. Omnisource achieved 96.4%, 71.7%, and 83.6% accuracy on the UCF101, HMDB51, and Kinetics400 dataset. PoseConv3D is a novel framework for HAR that relies on a 3D heatmap volume instead of a graph sequence as the base representation of human skeletons is traduced by Duan et al. [34]. The authors tested the proposed method on five skeleton benchmark datasets achieving an average accuracy of 97.1%, 90.3%, 47.7%, 94.3%, 76.7%, and 55.6% on the NTU60, NTU120, Kinetics, FineGYM, HMDB51, and UCF101datasets, respectively. Appendix A provides a summary of the various deep learning-based methods published in recent years, highlighting their performance across different modalities and datasets.

#### 5.2.2. Attention-Based Networks

Attention-based networks, particularly transformers, have emerged more recently as highly effective methods for human activity recognition (HAR) across a variety of sensing modalities, including RGB, mobile phones, wearable sensors, depth, and skeleton data. Originally developed for natural language processing, transformers have been adapted for sequential data modelling in HAR due to their ability to capture long-range dependencies and focus on relevant portions of the input sequence [239]. Unlike traditional recurrent architectures, transformers use attention mechanisms that allow the network to weigh different parts of the input based on their relevance, making them highly effective for tasks involving complex and multivariate sensor data [240]. The self-attention mechanism in transformers dynamically adjusts the importance assigned to each part of the input, enabling the model to focus on salient patterns and discriminative information crucial for recognising activities [239]. This ability is particularly useful in HAR systems, where human movements can exhibit complex temporal dependencies. In comparison to other architectures, attention-based models excel in their capacity to handle long-term dependencies without suffering from issues like vanishing gradients, which is a common problem in Recurrent Neural Networks (RNNs) and Long Short-Term Memory (LSTM) networks [241].

A Two-Stream Inflated 3D ConvNet (I3D) based on 2D ConvNet inflation is introduced by Carreira et al. [242] which used RGB and flow for action recognition on complex datasets such as the UCF-101, HMDB-51, and Kinects dataset with accuracy scores of 97.9%, 80.2%, and 78.7%. Girdhar et al. [243] proposed a Video Action Transformer Network, a novel architecture that localises and recognises actions in video clips by attending to the entire video sequence. The authors evaluated the performance of the model using frame-level mean average precision (frame-AP) at an intersection of union (IOU) threshold of 0.5.

Li et al. [244] introduced GroupFormer, a clustered spatial-temporal transformer network for group activity recognition. In the first step, the model extract features for individual people in a video and then applies a clustering algorithm to group them. Subsequently, the grouped features are fed into a transformer network that learns to capture the interactions between members in each group and predict the group activity. The model was pre-trained on Kinetics [242] using the I3D model and evaluated on the Volleyball dataset [72] and the Collective Activity dataset [245], and achieved an average accuracy of 95.7%, and 96.3%, respectively.

The Video Transformer Network (VTN), a transformer-based system for video recognition, is presented by Neimark et al. [246]. The VTN uses a 2D Convolutional Neural Network as a backbone to extract spatial features. Then, a transformer network is used to model the temporal relationships between frames, enabling the classification of actions by attending to the entire video sequence. The model is evaluated on the Kinetics-400 and Moments in Time datasets for video action recognition. The results show competitive accuracy while exhibiting significant speed advantages during training and inference compared to state-of-the-art methods.

VidTr, a video transformer network that recognises video actions without the use of convolutions, is presented in this study [247]. The network takes raw video frames as the input and uses a transformer architecture with separable spatial and temporal attention to learn action representations. Six widely used datasets are used to evaluate the model. The datasets are the Kinetics-400, Kinetics-700, Something-Something V2, Charades [103], UCF-101 [99], and HMDB-51 [248]. VidTr achieved a state-of-the-art performance on these datasets, demonstrating its effectiveness in capturing long-term temporal dependencies for video action recognition.

The method [240] proposed by authors investigates the use of self-attention mechanisms for video comprehension. It presents TimeSformer, a transformer-based architecture that applies different spatio-temporal attention mechanisms to capture interactions between video patches. The authors investigated various attention schemes, including divided space–time attention, joint space–time attention, and sparse local attention. A number of video action recognition benchmarks, including Kinetics-400, Kinetics-700, Something-Something V2, and HowTo100M are used to assess the performance of the proposed model. The TimeSformer achieved competitive results on these datasets, demonstrating the effectiveness of self-attention for video comprehension.

A video transformer model with space–time mixing attention that aimed for efficient video recognition is presented in the study [249]. The model restricts time attention to a local window and utilises efficient space–time mixing to attend to spatial and temporal locations without significant overhead. Lightweight global temporal-only attention mechanisms are integrated for further accuracy improvements. Experiments on Something-Something V2, Kinetics, and Epic Kitchens [122] datasets show high recognition accuracy and improved efficiency compared to other video transformer models.

Motionformer, a video transformer architecture that introduces a novel trajectory attention mechanism for video action recognition is proposed by the authors in reference [250]. The trajectory attention effectively captures the motion patterns in videos by aggregating data along implicitly determined motion trajectories. Motionformer was evaluated on three video-based action recognition datasets: Kinetics-400, Something-Something V2, and Epic Kitchens-100. The model achieved comparable results, demonstrating the effectiveness of the trajectory attention mechanism for video understanding.

This study [251] introduced Multiscale Vision Transformers (MViT), a transformer architecture for video and image recognition. The MViT model incorporated multiscale feature hierarchies, starting with high spatial resolution and low channel capacity in early layers, and progressively increasing the channel capacity while reducing spatial resolution in deeper layers. Both low-level and high-level visual information are effectively captured by this architecture. The model is evaluated on the Kinetics-400, Kinetics-600, Kinetics-700, Something-Something V2, and ImageNet datasets. MViT achieves state-of-the-art performance on these datasets, showcasing its effectiveness in video and image recognition tasks.

A pure transformer-based model for video classification titled ViViT is proposed by Arnab et al. [252]. By adapting the success of transformer models in image classification, the proposed model used them to handle the spatio-temporal nature of video data. From input videos, ViViT extracts spatio-temporal tokens and encodes them using a series of transformer layers. The proposed method used an efficient model variant that factorises the spatial and temporal dimensions of the input to address the challenge of long token sequences in videos. ViViT is evaluated on multiple video classification benchmarks, including Kinetics-400 and 600, Epic Kitchens, Something-Something V2, and Moments in Time. The model achieved state-of-the-art results on datasets evaluated which outperformed previous methods based on deep 3D convolutional networks.

A novel visual representation learning approach, namely the TokenLearner, using a few adaptively learned tokens for image and video recognition is presented in this study [253]. Instead of relying on pre-defined splitting strategies or densely sampled patches, TokenLearner learns to extract important tokens from the input data. This efficient tokenisation representation captures long-range interactions in videos and effectively models spatial contents in images. TokenLearner was evaluated on the ImageNet, Kinetics-400, Kinetics-600, Charades, and AViD [254] datasets. The model achieved strong performance on these benchmarks demonstrating the effectiveness of adaptive tokenisation for visual representation learning.

This study [255] proposed a shifted chunk transformer (SCT) for learning spatio-temporal representations in video-based action recognition. Image chunk self-attention blocks and shifted multi-head self-attention blocks are the two components of the SCT model. The image chunk self-attention block is used to capture local spatial dependencies within a chunk, while the shifted multi-head self-attention block captures long-range dependencies across different chunks. The model is evaluated on the Kinetics-400 and HMDB datasets. The SCT achieved competitive performance compared to existing methods, demonstrating its effectiveness for spatio-temporal representation learning in video action recognition.

Unary-pairwise transformer (UPT) which is a novel framework for human–object interaction (HOI) detection was introduced by the authors in reference [256]. The UPT processes each human–object pair individually and integrates unary and pairwise representations for enhanced interaction understanding. The UPT architecture incorporated a novel attention mechanism that attends to relevant regions in both the human and object features. The authors evaluated the model on two benchmark datasets, namely the HICO-DET dataset and V-COCO dataset. The results show that the UPT achieves state-of-the-art performance on both datasets.

Uncertainty-Guided Probabilistic Transformer (UGPT), another novel method introduced for complex action recognition using probabilistic modelling to quantify uncertainty in predictions, is introduced by the authors of [257]. The UGPT models attention scores as Gaussian random variables, capturing stochasticity in both data and predictions. The authors also proposed a novel training and inference strategy guided by this uncertainty, using majority and minority models to improve prediction accuracy and robustness. The method achieves state-of-the-art performance on the Breakfast Actions, MultiTHUMOS, and Charades datasets.

RegionViT [258], a novel vision transformer architecture, incorporated regional-to-local attention to enhance performance and efficiency in various computer vision tasks. Similar to Convolutional Neural Networks (CNNs), RegionViT utilised a pyramid structure to extract features at multiple scales. By limiting its scope while preserving the capability to capture global information, the regional-to-local attention mechanism reduced the computational burden of self-attention. The authors carried out experiments on multiple datasets including the ImageNet, CIFAR10, and CIFAR100 datasets. RegionViT achieved comparable performance across object detection and action recognition.

A novel video transformer architecture called RViT is introduced in this paper [241]. The proposed method used recurrent units to effectively process videos for action recognition. RViT incorporated a specifically designed attention gate mechanism to aggregate temporal attention recursively which processed both the current input frame and the hidden state from the previous frame, allowing the model to capture long-term temporal dependencies without being limited by video length. RViT operates on a frame-by-frame basis. The effectiveness of RViT is demonstrated through experiments on four benchmark datasets for human action recognition: Kinetics-400, Jester, Something-Something V2, and Charades.

DirecFormer is introduced in this work [259], which is also a transformer-based action recognition framework that incorporated a novel Directed Attention mechanism to address the issue of ordered temporal learning in videos. DirecFormer aimed to ensure that the model learns the correct order of actions in a video sequence, rather than being biased by background scenes or spurious correlations. The Directed Attention mechanism provides attention to human actions in the proper order, leading to improved robustness and generalisation. Experiments were carried out on the Jester, Something-Something V2, and Kinetics-400 datasets and DirecFormer achieved state-of-the-art results.

This paper [260] proposed UniFormer, a novel transformer architecture for efficient spatio-temporal representation learning in videos. UniFormer integrated 3D convolution and spatio-temporal self-attention within a unified transformer framework. The model utilised local multi-head relation aggregation (MHRA) in early layers to reduce computational complexity and global MHRA in deeper layers to capture long-range dependencies which addresses the redundancy and dependency issues inherent in video data. The model was evaluated on Kinetics-400/600 and Something-Something V1/V2. The results demonstrate that UniFormer achieves a good balance between accuracy and efficiency.

This paper [261] explored using image classifiers for the task of action recognition, challenging the traditional approach of using specialised temporal models. The methodology involves rearranging video frames into a “super image” based on a predefined spatial layout. This super image is then fed into an image classifier, treating action recognition as an image classification task. The paper focuses on action recognition and evaluates the approach on datasets such as Kinetics-400, Moments in Time, Something-Something V2, Jester, and Diving48. The authors experimented with Swin Transformer, a vision transformer model, and achieved state-of-the-art results, outperforming existing methods while being computationally efficient.

A novel Multiview Transformers approach to video recognition utilising multiple views of the input video to capture multi-resolution temporal context is presented in this work [262]. The model processed multiple temporal segments of the video in parallel, each at a different resolution, which effectively captured short-term and long-term temporal dependencies. This multiview strategy allowed the model to learn rich representations of the video content, leading to improved performance compared to single-view transformer architectures. The authors evaluated Multiview Transformers on six popular video classification datasets: Kinetics-400, Kinetics-600, Kinetics-700, Something-Something-V2, Moments in Time, and Epic Kitchens.

A self-supervised learning approach called Self-supervised Video Transformer (SVT) for training video transformers on unlabelled video data is presented in this study [239]. This method involves creating local and global spatio-temporal views of a video with varying spatial sizes and frame rates. A self-supervised objective is then used to match features from these diverse views, ensuring invariance to spatio-temporal variations in actions. The effectiveness of SVT is evaluated on four action recognition benchmarks: Kinetics-400, UCF-101, HMDB-51, and SSv2. The results show strong performance, particularly the ability to converge quickly with small batch sizes.

VideoMAE [263] is another a self-supervised video pre-training (SSVP) method based on masked autoencoders, specifically designed for video transformers. It employed an asymmetric encoder-decoder architecture where a high proportion of video cubes are masked, and the model is tasked with reconstructing these missing cubes. VideoMAE is evaluated on datasets such as Kinetics-400, Something-Something V2, UCF101, and HMDB51. The method achieves impressive results, particularly in data-efficient settings.

An extension of Masked Autoencoders (MAEs) for learning spatio-temporal representations from videos is presented in this work called the ST-MAE [264]. The approach, similar to ImageMAE, involves randomly masking out spacetime patches in videos and training an autoencoder to reconstruct them in pixels. The method is evaluated on action recognition benchmarks, including Kinetics-400, Kinetics-600, and Something-Something V1 and V2. The results demonstrate superior performance over supervised pre-training, achieving, for example, 82.9% top 1% accuracy on Kinetics-400 with ImageNet-1K pretraining.

Long-Short Temporal Contrastive Learning (LSTCL), a self-supervised pre-training technique for video transformers that leverages the models’ capability to capture long-range dependencies, is introduced in this work [265]. LSTCL creates positive pairs by contrasting the representation of a short video clip with that of a longer clip from the same video. The method encourages the model to learn clip-level representations that incorporate a broader temporal context. This paper focused on action recognition and evaluated LSTCL on datasets such as Kinetics-400, Kinetics-600, SomethingSomething V2, UCF101, and HMDB51, demonstrating performance on par with or surpassing supervised pre-training on large image datasets.

A novel approach for pre-training video transformers using a BERT-style objective named BEVT is introduced in this work [266], aiming to learn discriminative video representations. BEVT utilised a two-stream network, separating spatial representation learning and temporal dynamics learning, allowing the model to effectively capture both types of features. The authors investigated the importance of spatial and temporal clues in video recognition, highlighting their varying influence across different video samples. BEVT is evaluated on three challenging video benchmarks: Kinetics-400, Something-Something V2, and Diving48 and achieved satisfactory scores.

OmniMAE, a novel masked autoencoder framework for single model pre-training on images and videos, is proposed by Girdhar et al. [267]. OmniMAE utilised a unified architecture and training procedure for learning representations across multiple visual modalities that facilitated efficient knowledge transfer and generalisation for activity learning. The model leverages the complementary nature of image and video data by jointly training on ImageNet (IN1K) and Something-Something V2 (SSv2) datasets. This method allows the model to capture a wide range of visual concepts, including static appearance and dynamic motion patterns. The effectiveness of OmniMAE is demonstrated through transfer learning experiments on image datasets (iNaturalist-2018 and Places-365) and video datasets such as Kinetics-400 and EPIC-Kitchens-100.

This paper [268] introduced Masked Motion Encoding (MME) which is a novel self-supervised learning framework for video representation learning. This method masks out portions of motion trajectories within videos and tasks the model with predicting the missing motion information. This approach encourages the model to learn robust and discriminative representations of motion patterns, which are essential for various video understanding tasks. The effectiveness of MME is evaluated through transfer learning experiments on action recognition benchmarks, such as Kinetics-400 and Something-Something V2. The results demonstrate that MME pre-training significantly improves the performance of action recognition models compared to other self-supervised approaches.

A motion-guided masking algorithm (MGM) to enhance the effectiveness of masked autoencoders (MAE) for spatio-temporal representation learning in videos is suggested by Fan et al. [269]. Instead of random masking strategies used in image MAEs, MGM used motion vectors obtained from compressed video formats to dynamically mask spatio-temporally salient regions in videos. This method was evaluated on Kinetics-400 and Something-Something V2 datasets and showed significant improvements over existing video MAE approaches. Furthermore, MGM exhibits faster convergence and superior generalisation capabilities in downstream transfer learning and domain adaptation tasks on the UCF101, HMDB51, and Diving48 datasets.

One key advantage of attention-based networks is their versatility across different sensor modalities [258]. By dynamically attending to relevant sensor readings or visual features, transformers can effectively integrate information from diverse sources such as combining RGB camera data with depth and skeletal readings to achieve a more comprehensive understanding of human activities [269]. This multimodal fusion enhances recognition accuracy and robustness in real-world HAR applications, making transformers particularly well-suited for complex environments, including healthcare monitoring, autonomous driving, and human–computer interaction.

As shown above, recent research in attention-based models has led to several innovations in architecture design. Variants such as the Vision Transformer (ViT), TimeSformer, Video Swin Transformer (Swin-T), masked autoencoders (MAE), Masked Motion Encoding (MME), and motion-guided masking algorithm (MGM) have demonstrated exceptional performance in HAR tasks, particularly in video-based human activity datasets. These models have shown superior generalisation capabilities and scalability, making them an attractive choice for HAR systems that require real-time processing and the ability to adapt to various contexts. In addition to exploring new attention mechanisms, researchers are also investigating the benefits of transfer learning to leverage pre-trained models and fine-tune them for specific HAR tasks. By building on large, pre-existing datasets, HAR systems using transformers can achieve faster convergence and improved performance, even when dealing with smaller, domain-specific datasets. This makes transformers an attractive option for applications where collecting large-scale labelled data is challenging. Ongoing efforts are focused on optimising transformer architectures for computational efficiency, as transformers are known for their high computational and memory demands. Methods such as efficient attention mechanisms and lightweight transformer variants aim to make these models more accessible for real-time and resource-constrained HAR applications. Appendix A summarises recent transformer-based models for HAR, showcasing their performance on popular datasets like Kinetics-400 (K400), UCF101, HMDB51, and Something-Something V2 (SSV2).

#### 5.2.3. Reinforcement Learning-Based Networks

Human activity recognition (HAR) is a complex research area that focuses on the identification and recognition of movements or activities performed by individuals. Although people can quickly identify actions they see in videos, fully automating this process is challenging, yet essential given its wide range of real-world purposes [270]. HAR is presented as a learning problem in the literature on computer vision, and machine learning techniques are used to categorise and identify the activities. Prior to the introduction of deep learning (DL) models in activity detection methods, researchers primarily focused on using manually extracted features such as motion descriptors and interest points [271,272] for HAR. DL techniques were first applied to the field to extract efficient video representations and increase generalisation for video-based HAR as hand-crafted feature extraction is time-consuming and requires extensive domain knowledge. This led to a notable advancement in the field.

In contrast to supervised learning techniques, reinforcement learning (RL) is a branch of machine learning techniques that learns in an unsupervised manner by having one or more agents interact with the surroundings [273]. Depending on a pre-determined reward function, these interactions produce either positive or negative rewards. Without being aware of the ground truth, the agent learns how to maximise the predicted reward through trial and error as it acquires additional information from the environment. Deep Reinforcement Learning (DRL), a novel family of algorithms whose efficacy has been shown in numerous applications, was created more recently by combining reinforcement learning with deep learning [274,275]. Reinforcement Learning (RL) and Deep Reinforcement Learning (DRL) methods represent a growing frontier in human activity recognition (HAR) [270], offering innovative ways to address dynamic and complex activity recognition tasks across various sensing modalities, including RGB, mobile phones, wearable sensors, depth, and skeleton data. In contrast to traditional supervised learning models, that depend on fixed datasets for training, RL-based methods learn optimal decision-making policies by interacting with an environment [270]. This adaptability makes RL particularly well-suited for HAR scenarios where human activities are continually evolving, and where real-time feedback can help improve model performance.

In RL-based HAR, the system is modelled as an agent that learns to recognise activities by taking actions and observing outcomes. The agent receives rewards or penalties applied based on the accuracy of its predictions [273]. Over time, the agent learns to optimise its activity recognition strategy through a trial-and-error process by making the RL-based method a promising approach for tasks that involve sequential decision-making. It is particularly beneficial for situations such as continuous monitoring of activities and autonomous systems. RL-based methods can be effective for real-time interventions in healthcare settings [274]. Deep Reinforcement Learning (DRL) which combines RL and deep learning methods extends this paradigm by using deep neural networks to handle high-dimensional data inputs, such as video streams or multivariate sensor data, which are commonly used in HAR systems. DRL models are particularly effective in HAR applications that analyse complex spatio-temporal patterns [270]. DRL methods can be applied to recognise activities in crowded environments where existing methods may fail. DRL methods adapt to changes in sensor configurations and learn new activities over time without manual intervention which make it a powerful tool for enhancing the robustness of HAR systems.

A spatio-temporal deep Q-network (ST-DQN) for human activity localisation is presented by the authors in reference [276], which consist of recognising activity category labels by identifying their spatio-temporal locations in video sequences. The model is an integrated framework that uses temporal dependencies and contextual information to improve localisation accuracy while concurrently addressing spatial and temporal localisation. With a limited number of proposals, the ST-DQN achieves promising localisation performance. By incorporating temporal dependencies, it outperforms previous frame-based localisation methods, achieving higher mean average precision (mAP) scores across different intersection-over-union (IoU) thresholds on challenging datasets like UCF-Sports [277], sub-JHMDB [278], and UCF-101 [99].

Deep progressive reinforcement learning (DPRL) which is a new technique for skeleton-based action recognition, is suggested by the authors’ research presented in reference [279]. The proposed method concentrated on selecting the most relevant frames from a skeleton sequence while it eliminated ambiguous frames that effectively increased action accuracy. The authors used deep reinforcement learning to model frame selection as a progressive process. The quality of the nominated frames for action recognition and their association to the entire video sequence were the two main factors that the DPRL method used to iteratively update the frame selection. Additionally, the study employed a Graph-based Convolutional Neural Network (GCNN) that captured the inherent graph-like structure of human body joints. In GCNN, vertices represent joints and edges depict the relationships between them. The DPRL method achieved 79.6%, 98.5, and 89.8% accuracy on the SYSU-3D [280], UT-Kinect [64], and NTU RGB+D 60 [105] datasets, respectively, which were used for skeleton-based action recognition.

This study [281] proposed a multi-agent reinforcement learning (MARL) framework for untrimmed video-based action recognition. The framework focused on developing a learning-based frame sampling strategy to enhance both the accuracy and efficiency of video classification tasks, especially for untrimmed videos where frame-level saliency varies significantly. The paper considered frame sampling as numerous simultaneous Markov decision processes, each of which is tasked with identifying an informative frame or clip by improving an initial sample point, as an alternative to hand-crafted sampling techniques. The authors [281] employed a novel RNN-based context-aware observation network to model contextual information between nearby agents and incorporate historical states of individual agents. The MARL is comprised of this network, along with a policy network that determines action probabilities and a classification network for reward calculation and final recognition. The authors used ActivityNet [101] v1.2 and v1.3 for an action recognition task achieving a mAP score of 64.13% and 62.00%, respectively.

A novel framework for effective human activity classification in egocentric videos which focused on reducing computational costs while maintaining accuracy is introduced in this study [282]. The approach combined actor-critic reinforcement learning (RL) with a deep-shallow network architecture to identify and process regions of interest (ROI) within video frames. The scope of the work is to efficiently classify human activities captured from a first-person perspective using wearable cameras. The authors evaluated their approach on the Dataset of Multimodal Semantic Egocentric Video (DoM-SEV), focusing on five activities, which are walking, running, standing, being in conversation, and browsing. In comparison to a traditional 3D CNN (C3D), the results demonstrated a 36.4% reduction in processing time while achieving similar accuracy.

A unique method for human activity recognition (HAR) that uses a camera mounted on a mobile robot is presented in the author’s work [283]. This research aimed to improve HAR accuracy by optimising the robot’s position and viewing angle through Deep Reinforcement Learning (DRL) while reducing energy consumption related to the robot’s movement. The key idea was to train the robot to dynamically adjust the position of the robot based on the observed activity and its confidence in recognising the activity. The study used a virtual environment called the HoME Platform [284], to evaluate the proposed method. Twelve activities were conducted by virtual actors in the simulated environment and subsequently used for classification which achieved 75% accuracy.

A deep learning framework [285] for location- and person-independent activity recognition using WiFi signals was suggested in the work. The objective of the work was to develop a robust system that recognises activities accurately regardless of the position and orientation of WiFi devices or the identity of the person who is performing the activity. Given that WiFi signals are highly sensitive to environmental factors and individual variations, The proposed method addressed a key challenge in WiFi-based activity recognition. The authors used three Deep Neural Networks (DNNs), to recognise activities that are location- and person-independent. Firstly, a 2D CNN was used to extract location- and person-independent features from different perspectives of the CSI data. Secondly, 1D CNN captured temporal dependencies between successive CSI segments. Finally, a neural architecture search combined RNN with LSTM which acts as a reinforcement learning agent, responsible for optimising the neural architecture of the recognition algorithm. The Yousefi-2017 [286] and FallDeFi [287] datasets were used to evaluate the performance of the proposed method.

In order to identify action classes that have not been observed during training, this study [288] introduced CLASTER, a novel approach for zero-shot action recognition. The authors suggest that generalisation to unseen action categories may be hampered by the precise decision boundaries that typical neural networks acquire in supervised environments. To overcome this, CLASTER makes use of a cluster-based representation that encourages the formation of a representation that successfully generalises to new action cases while also regularising the learning process. Instead of using individual data points, CLASTER aims to learn a more general representation that is less susceptible to the variations present in individual instances by using cluster centroids. Reinforcement Learning (RL), more specifically the REINFORCE algorithm, is used to optimise the clustering. This allows direct optimisation of the cluster centroids based on the classification performance. The UCF101 [99], HMDB51 [248], and Olympic Sports [289] datasets were used for performance evaluation.

One of the key advantages of RL and DRL in HAR is their ability to learn from interactions with an environment, enabling systems to adjust to new activity patterns or sensor inputs without the need for extensive retraining. This makes RL-based HAR systems highly adaptable and scalable across different environments and applications, such as smart homes, autonomous driving, and healthcare monitoring. However, there are several challenges associated with RL-based HAR. The exploration of large state-action spaces in complex environments can require significant computational resources and time. Additionally, ensuring safety during the learning process is crucial, especially in applications like healthcare or autonomous systems, where incorrect decisions could have serious consequences. To address these issues, researchers are exploring various techniques, including reward shaping, transfer learning, and simulation-based training environments, to improve the efficiency and safety of RL-based HAR models. Despite these challenges, RL and DRL are gaining traction in HAR research, with ongoing developments in algorithmic efficiency, policy optimisation, and hybrid methods that combine RL with other machine learning paradigms to enhance performance. As RL techniques continue to evolve, they are expected to play a more prominent role in creating adaptive and autonomous HAR systems capable of handling complex, real-world scenarios. A summary of HAR methods using RL and DRL-based techniques is presented in Appendix A.

### 5.3. Machine Learning Algorithm Performance Comparison

Comparing the performance of various machine learning algorithms in human activity recognition (HAR) is crucial to identify the most effective approaches for specific application scenarios. The selection of algorithms often depends on factors such as accuracy, computational complexity, and scalability, which vary depending on the data modality and the target application. Each machine learning method brings its own strengths and limitations in addressing the diverse and dynamic nature of human activities.

Supervised learning algorithms like Support Vector Machines (SVM), Decision Trees (DT), Random Forests (RF), and K-Nearest Neighbours (K-NN) often achieve high accuracy, especially when most effective features are used and trained on large, well-labelled datasets, particularly in environments with clearly defined activity patterns as shown in the investigation [76], but these algorithms may encounter challenges when applied to recognising more complex, multi-step activities [290]. In contrast, semi-supervised learning techniques, which combine both labelled and unlabelled data, offer an effective alternative in scenarios where labelled data are scarce. However, these methods generally perform slightly lower than fully supervised approaches due to the challenges of effectively utilising unlabelled data [290].

On the other hand, deep learning methods, including CNNs and RNNs, have consistently outperformed traditional approaches due to their ability to automatically extract meaningful features from raw data, such as images or acceleration [291]. CNNs excel in processing spatial data like RGB images, while RNNs are particularly suited for temporal sequences, such as time-series data from wearable sensors, or skeletal joint movement data allowing superior activity recognition. Convolutional Neural Networks (CNNs) and Recurrent Neural Networks (RNNs) represent a significant advancement in HAR [215]. By automatically learning hierarchical and discriminative features from raw sensor data, these models achieve superior accuracy and generalisation across complex datasets. Attention-based networks and transformers further enhance accuracy by focusing on the most critical features within input sequences [292] by improving both interpretability and recognition performance [293]. By dynamically weighting relevant information, these networks improve the model’s ability to recognise complex activities with greater precision.

Reinforcement Learning (RL) methods, while less common in HAR, offer a unique advantage by enabling adaptive decision-making based on feedback from the environment [285]. Although Reinforcement Learning (RL) methods can also achieve high accuracy, they often require extensive exploration to optimise performance. RL-based approaches require extensive exploration and careful reward design, as they may present safety concerns in real-world applications where incorrect decisions could have serious consequences. RL-based HAR systems iteratively improve over time but may take longer to reach peak accuracy compared to supervised or deep learning models. To evaluate the performance of HAR methods, different metrics are used. In the section below, various evaluation metrics that are found in the literature used by authors to evaluate different methods are described.

#### 5.3.1. Evaluation Metrics

In the context of human activity recognition (HAR), the terms True Positive ($T_{p}$), False Positive ($F_{p}$), False Negative ($F_{n}$), and True Negative ($T_{n}$) are used to evaluate the performance of a classification model. True Positives refer to the number of times an HAR model correctly predicts the activity as the correct class. False Positive indicates the number of times a model incorrectly classifies an activity as a certain class when it was actually a different activity [294]. False Negative signifies the number of times the method has failed to predict the correct activity and instead predicted a different activity. True Negatives denote the number of times an HAR classifier correctly identified that a certain activity did not occur. In the section below, commonly used metrics for HAR evaluation are listed with brief description.

Accuracy

As seen in the literature [41,190], accuracy is a widely used metric for evaluating the effectiveness of machine learning models which are also adapted to assess the performance of human activity recognition (HAR) systems. In the context of HAR, accuracy refers to the proportion of correctly identified activity samples compared to the total number of activity samples, across all classes and classes. Accuracy is defined as [294](1)$$Accuracy=\frac{Number\;of\;Correct\;Predictions}{Total\;Number\;of\;Predictions}=\frac{T_{p}+T_{n}}{T_{p}+T_{n}+F_{p}+F_{n}}$$
where $T_{p}$, $T_{n}$, $F_{p}$, and $F_{n}$ are True Positive and Negative, and False Positive and Negative, respectively.

Precision

Precision quantities the relationship of properly identified positive sample (True Positives) out of all the positive predictions, reflecting HAR model’s ability to minimize False Positives. Precision is defined as [294](2)$$Precision=\frac{T_{p}}{T_{p}+F_{p}}$$

Recall (Sensitivity)

Recall or sensitivity measures the ratio of True Positive predictions out of all actual positive instances in the dataset. Recall is defined as [294](3)$$Precision=\frac{T_{p}}{T_{p}+F_{n}}$$

F1-Score:

F1-Score measures the harmonic mean of precision and recall, providing a balance between the two. It is particularly useful when the classes are imbalanced. F1-Score is defined as [294](4)$$F1-Score=2\times \frac{Precision\times Recall}{Precision+Recall}$$

Confusion matrix:

A confusion matrix or error matrix is a table that provides a summary of the performance of a classification algorithm by showing the error made by the classifier including the True Positive, True Negative, False Positive, and False Negative values.

Intersection-over-Union (IoU):

IoU, also called Jaccard index, determines the accuracy of the detector on a specific dataset. When classifying human activity using sensors or video data, IoU can be treated as the overlap between predicted activity intervals (the time where a particular activity was recognised) and true activity intervals (the actual time when the activity occurred). IoU is defined as [295](5)$$IoU=\frac{Area\;of\;overlap}{Area\;of\;union}$$

Mean Average Precision (mAP):

In the context of information retrieval, object detection, and multi-label classification tasks, mean average precision (mAP) is a commonly used evaluation metric and it is also applied to evaluate vision-based HAR systems. Average Precision (AP) refers to the average precision scores estimated at different recall levels for a particular activity class. Average Precision is defined as [295](6)$$AP=\frac{1}{N}\sum_{i=1}^{N}Precision\;(r_{i})$$
where *N* is the number of recall levels, and Precision ($r_{i}$) is the precision at recall level $r_{i}$.

Receiver Operating Characteristic (ROC) and Area Under the Curve (AUC):

ROC is popular graphical evaluation metric used in machine learning which is also applied in the evaluation of HAR systems. True Positive Rate (TPR) which measures the percentage of actual positive instances that are correctly identified by the model and False Positive Rate (FPR) which measures the amount of actual negative instances that are incorrectly classified as positive by the model are used to plot the ROC curve. Typically, TPR is plotted in the *X*-axis and FPR is plotted in the X-axis. TPR and FPR are calculated as [296](7)$$TPR=\frac{T_{p}}{T_{p}+F_{n}}$$(8)$$FPR=\frac{F_{p}}{F_{p}+T_{n}}$$

The Area Under the Curve (AUC) refers to the area under the ROC curve which provides a single scalar value between 0 and 1 to summarise the performance of a classification model across all possible thresholds. The model’s ability to differentiate between positive and negative classes is measured by its AUC.

Internal Cluster Evaluation Metrics

When class labels are not available for the HAR dataset, typically unsupervised learning methods are applied to uncover the patterns within the data, so clustering methods such as K-means, Spectral, Hierarchical, and density-based clustering are often applied for that purpose [297]. Internal metrics assess the quality of clustering based purely on the clustering results themselves without using ground truth labels [83].

Silhouette Score

Silhouette Score is a frequently used matrix that evaluates a data point’s similarity to its own cluster in relation to other clusters. It defined as [137](9)$$Silhouette\;score=\frac{b-a\;}{\;max(a,b)}$$
where a represents the mean distance between a data point and every other data point within the same cluster and b is the average distance from the point to all points in the nearest cluster.

Davies–Bouldin Index (DBI)

The Davies–Bouldin Index (DBI) calculates the average similarity ratio between each cluster and the cluster that is most comparable to it; typically, a lower DBI indicates better clustering. DBI is calculated as [137](10)$$DBI=\frac{1}{k}\;\sum_{i=1}^{K}max\frac{\sigma_{i}+\sigma_{j}}{d_{ij}}\;where\;i\neq j$$
where *K* is the number of clusters. $\sigma_{i}$ is the average distance between samples in cluster *i*. $d_{ij}$ is the distance between the centroids of clusters *i* and *j*.

Dunn Index

The minimal inter-cluster distance divided by the maximum intra-cluster distance is measured by the Dunn Index. Better clustering is indicated by a higher Dunn Index. The Dunn Index is given by [137](11)$$Dunn\;Inedx=\frac{{min}_{i\neq j}distance(C_{i},C_{j})}{{max}_{k}distance(C_{k})}$$
where $C_{i},C_{j}$ are clusters *i* and *j*. $distance(C_{k})$ is the maximum distance between the points in cluster k.

External Evaluation Metrics

The clustering results are compared with ground truth labels (where available) using external measures. Expert-labelled activity data may be used to establish ground truth labels in unsupervised HAR, or heuristics or assumptions may be used to approximate them in the absence of expert labelling.

Adjusted Rand Index (ARI):

ARI evaluates the similarity of two clustering results to one another (e.g., unsupervised clusters vs. ground truth). It corrects for chance, which means it takes into consideration the possibility that some apparent resemblance could result from random clustering. ARI ranges between −1 and 1: 1 means perfect clustering, 0 means clustering is better than random clustering, while −1 denotes the worst clustering. ARI is defined as [137](12)$$ARI=\frac{RI-E[RI]}{\max\left(RI\right)-E[RI]}$$
where RI is the Rand Index and E[RI] is the Expected Rand Index.

Normalised Mutual Information (NMI):

NMI evaluates the quantity of information that is shared between the ground truth and the clustering results. To guarantee that the outcome falls between 0 (no mutual information) and 1 (perfect agreement), it is normalised. It is defined as [137](13)$$NMI=\frac{I(U,V)}{\sqrt{H(U)H(V)}}$$
where the mutual information between the clustering and the ground truth is denoted by I(U, V). H(U) and H(V) represents the entropies of the clustering and the ground truth, respectively.

Fowlkes–Mallows Index (FMI):

The geometric average of the pairwise precision and recall between the ground truth and clustering is measured by FMI. Typically, it ranges between 0 and 1, with 1 signifying ideal clustering and values close to 0 mean bad clustering. FMI is calculated as [137](14)$$FMI=\frac{T_{p}}{\sqrt{{(T}_{p}+F_{p})(T_{p}+F_{n})}}$$

#### 5.3.2. Computational Complexity

Computational complexity refers to the number of computational resources—such as time, memory, and processing power—required to train and deploy machine learning algorithms for human activity recognition (HAR). This factor is particularly important in real-time or resource-constrained environments where processing must occur quickly and efficiently [282]. Machine learning models, such as SVM, KNN, and Decision Trees, tend to have lower computational complexity compared to deep learning methods. These models are generally faster to train and deploy, making them well-suited for applications where real-time processing or limited computational resources are a concern. However, as datasets grow in size or dimensionality, the time required to train these models can increase significantly, potentially limiting their scalability for large-scale HAR tasks. In contrast, deep learning models, including CNNs and RNNs, are computationally much more demanding. These models involve multiple layers and a large number of parameters, which makes training and inference resource-intensive [282]. Deep learning models often require GPUs or cloud-based resources to manage the high computational load, particularly when handling large-scale HAR datasets or complex activity patterns.

Attention-based methods, such as transformers, further increase computational complexity [282]. These methods calculate attention weights for each element in the input sequence, which can slow down both training and inference. Although more efficient attention mechanisms have been introduced in recent transformer architectures, attention-based models still require significant computational power to function effectively [282]. Reinforcement Learning (RL) methods are often the most computationally complex. RL models must explore large state-action spaces to learn optimal strategies for recognising human activities, a process that is both time-consuming and resource-intensive [281]. Training an RL agent in environments with high-dimensional data or complex activity patterns often demands substantial computational resources, making RL methods challenging to implement in real-time HAR systems without sufficient computational power. Methods such as AttentionClusters [281] may be explored for reducing complexity.

#### 5.3.3. Scalability and Generalisation

Scalability refers to the ability of a machine learning algorithm to efficiently handle increasing amounts of data, complexity, or computational resources without significant performance degradation [298]. In the context of human activity recognition (HAR), scalability determines whether a model can adapt and perform effectively as the size of datasets, number of activity classes, or complexity of sensor environments grows [187].

Supervised learning algorithms, while typically effective for smaller datasets or less complex HAR tasks, may struggle with scalability when applied to larger or more diverse datasets [299]. As data volume increases, the time and computational resources required for training these models can grow exponentially. However, advancements in parallel processing and distributed computing, particularly in cloud environments, have enabled supervised learning algorithms to scale more effectively, making them viable for large-scale HAR applications [297]. Deep learning models, particularly Convolutional Neural Networks (CNNs) and Recurrent Neural Networks (RNNs), are designed to scale well with large datasets due to their ability to learn complex hierarchical representations [291]. Their scalability allows them to capture intricate patterns in human activities from vast amounts of data, such as video sequences or time-series sensor data. However, this scalability comes at the cost of substantial computational resource requirements, limiting their deployment in environments with constrained hardware. Techniques like transfer learning [299] and model compression are increasingly used to improve the scalability of deep learning models, allowing them to function in resource-limited settings without significantly compromising accuracy.

Attention-based networks, such as transformers, can scale to larger datasets as the amount of the data and model grow, but they often encounter issues related to interpretability and increased computational complexity [292], making it harder to understand how the model is making decisions. Efforts are underway to optimise these architectures for larger-scale HAR applications, focusing on more efficient attention mechanisms that can handle increased data complexity without incurring excessive computational costs. Reinforcement Learning (RL) presents unique scalability challenges in HAR, particularly when the state-action space becomes too large [281]. RL algorithms must explore vast environments or datasets to learn optimal decision-making policies, which can make them computationally expensive and time-consuming to train. However, RL models can scale through efficient exploration and optimisation strategies [281]. In practice, this makes them more suitable for controlled environments or applications with well-defined activity patterns, as their scalability is often limited by the complexity of the environment and the availability of computational resources for extensive exploration [282].

## 6. Applications of Human Activity Recognition

Across a wide range of domains, human activity recognition (HAR) technology has proven its versatility and effectiveness turning it into a crucial tool for addressing various real-world problems [300]. The ability to effectively monitor, analyse, and recognise human actions using data from different types of sensors has opened opportunities for improving human lives. Fields such as safety, performance optimisation, enhancing everyday living [62], healthcare [301], sports tracking and analysis [302], security and surveillance, and smart environments [303] are among the many sectors where HAR has found practical applications.

Successful implementations of HAR systems across these domains showcase the broad applicability of this technology. For example, HAR systems can significantly improve fall detection and provide immediate response [304] in places like elderly care or individuals who are living alone. In sports, HAR can be used for performance monitoring and enhanced athletic training programs to facilitate training and prevent injury. Similarly, HAR systems can be deployed for intrusion detection and behaviour monitoring in public spaces to strengthened security measures. These are examples of real-world applications of HAR that emphasise the potential of the technology across a variety of fields, where it continues to advance rapidly.

The applications discussed above are not exhaustive. Researchers are continuously exploring new areas where HAR can be applied, especially as advancements in sensing modalities, learning algorithms, and computational power allow complex activity to be recognised with improved accuracy. From autonomous vehicles to workplace safety and beyond, HAR is poised to expand its impact, fostering innovative solutions to emerging challenges across a multitude of industries. In the section below, some of the HAR application are briefly described.

## 7. Challenges and Future Directions

As human activity recognition (HAR) continues to evolve, the research landscape is increasingly shaped by emerging trends and ongoing advancements in this field [91]. All the upcoming technologies such as multimodal sensor fusion, edge computing, IoT-based systems, and personalised adaptive models are becoming pivotal in expanding the scope and effectiveness of HAR applications. The increase in focus towards growing attention to ethical and social considerations are driving HAR towards more user-centric solutions.

However, the field faces several challenges that must be addressed for HAR systems to perform optimally. Technical hurdles introduced by sensors, limited availability of data segmentation methods, and model interpretability and transparency are needed for that purpose. Ensuring that HAR models can be generalised across diverse environments and transferred to new applications is crucial for scalable implementation. Moreover, human-centric design and evaluation frameworks are necessary to ensure HAR technologies align with user needs and societal expectations. The future of HAR research lies in addressing these challenges while embracing technological advancements, enabling more efficient, ethical, and human-centred systems.

### 7.1. Emerging Trends and Future Directions in Human Activity Recognition Research

Researchers are increasingly exploring innovative approaches to tackle current challenges and unlock new opportunities in human activity recognition (HAR). Future directions in HAR are focused on enhancing its accuracy and applicability across a variety of contexts. These approaches also take into consideration the ethical implications of widespread adoption.

One notable advancement within this domain is that of multimodal sensor fusion, in which HAR systems utilise data from different sensor modalities, such as RGB cameras or inertial and environmental sensors, to gain a better understanding of human activities [305,306,307]. Traditional HAR systems like the sensor-based and vision-based systems often face limitations due to factors like occlusion. Changes in lighting condition and environmental variability leads to lower accuracy. By combining data from multiple sensors, future HAR systems may offer improved accuracy and increase the reliability of HAR systems. The fusion of sensors are expected to overcome some of the existing challenges and allow fine-grained activity recognition across a variety of settings [308,309]. Multimodal sensor fusion techniques facilitate comprehensive analysis of human actions that capture complementary data from different sources to enhance the overall performance of HAR systems [309].

Another emerging trend is the integration of edge computing and Internet of Things (IoT)-based systems. These technologies capture and process sensor data directly at the edge of the network. Edge devices reduce latency and improve privacy as they do not need to send data to any centralised servers [310]. This is particularly useful in HAR applications that require real-time decision-making, such as healthcare monitoring or smart home systems. Edge computing also helps to conserve computational resources, leading to more scalable and efficient HAR deployments in IoT environments [310].

Adaptive models represent another area of growth in HAR research. Traditional HAR models often struggle with generalisation as they are designed to identify activities based on a fixed dataset. Those fixed models do not consider the variations of individual differences in activities. Future research aims to develop models that can adapt to the unique habits, preferences, and contexts of individual users [311,312]. By incorporating techniques such as reinforcement learning and user feedback, these personalised models may offer user-specific activity recognition that dynamically adjust to changes in activities and environments [214].

Ethical and social considerations are becoming increasingly important as HAR systems are gradually integrated into daily life. As these systems process sensitive data about human actions, privacy concerns and data security comes forward. Algorithmic bias and transparency of the model interoperability have come to the forefront [313]. Future HAR research will need to prioritise these issues to ensure that the technology is deployed responsibly considering data safety that protects the privacy of users [314]. Addressing these ethical and social challenges is essential for building trust to ensure the widespread adoption and acceptance of HAR technologies in various sectors.

These trends illustrate how future HAR research is moving towards creating more accurate, adaptable, and ethically responsible systems that can be applied across a broad spectrum of real-world scenarios.

### 7.2. Research Challenges and Potential Solutions

Despite significant advances in human activity recognition (HAR), numerous challenges persist that hinder the full realisation of its potential, particularly as HAR applications increasingly leverage artificial intelligence (AI), machine learning (ML), and deep learning (DL) methods. A wide range of obstacles, such as intraclass variations, view changes, meaningful feature extraction, the availability of large-scale datasets, multi-occupant activity recognition, computational costs, and privacy concerns, continue to pose challenges to the field, as identified in various surveys [315].

One prominent challenge is related to HAR modalities. Vision-based approaches, while the least intrusive, are sensitive to environmental factors and equipment limitations [305]. For instance, RGB camera-based HAR methods provide rich visual information but are highly susceptible to issues such as illumination variations and view changes, making it difficult to maintain consistent recognition accuracy across different environments. Additional challenges arise from intraclass variations, where activities that should be recognised as the same category exhibit different characteristics (e.g., falls that differ in their visual representation), self-occlusion, and the need to recognise activities that are differentiated by subtle movements. These challenges are illustrated in Figure 9, showing examples of intraclass variations of fall (Figure 9a–e).

Viewpoint variations are illustrated in Figure 10a,b, showing a person sitting from different angles, highlighting the impact of changes in viewing perspective on human activity recognition (HAR) accuracy. An example of synchronised activity is shown in Figure 10c, where a person is relaxing on the couch and reading a book, demonstrating how concurrent actions can complicate activity classification. Subtle differences in motion, such as running versus jogging, are depicted in Figure 10c and 10d, respectively, showcasing the challenge of distinguishing activities that only vary slightly in movement patterns. Lastly, Figure 10f illustrates the issue of self-occlusion, where parts of the body, like limbs or joints, are obscured by other body segments, complicating the recognition process [46].

Depth sensors, such as Microsoft Kinect, address challenges related to varying lighting conditions but come with inherent limitations, including a restricted field of view and limited portability, which constrain their application in more dynamic or expansive environments [316]. Motion capture (Mocap) systems, while providing highly accurate and detailed data suitable for controlled environments like those used in sports injury rehabilitation or entertainment, are often prohibitively expensive and impractical for everyday activity monitoring. In contrast, wearable sensors offer a robust, view-invariant approach to HAR, capable of recognising movements regardless of environmental factors. However, these sensors typically recognise only a limited range of activities and may cause user discomfort or even skin irritation when worn for extended periods, as demonstrated in Figure 11a,b.

Data segmentation and labelling represent another major challenge for HAR research [127]. The process of segmenting continuous data streams and accurately labelling activities is time-consuming, labour-intensive, and prone to errors, particularly for complex or overlapping activities and across diverse sensor modalities [305]. Various existing windowing methods commonly used by HAR researchers are portrayed in Figure 12. Manual annotation of data is often required, introducing inconsistencies and limiting scalability. In response, potential solutions include crowdsourcing annotation tasks to distribute the workload across a larger pool of participants, as well as developing automated and semi-automated annotation tools. The authors in reference [127] have proposed an autoencoder-based approach that use frame reconstruction errors to identify activity transition points. Though the method is intuitive, it requires few hyperparameter initialisations. More research in this field is required to fully automate the activity segmentation process. Additionally, transfer learning techniques can be used to adapt pre-trained models from similar tasks or domains, reducing the reliance on large manually labelled datasets while maintaining model accuracy and relevance.

Another critical issue lies in model interpretability and transparency [317]. As deep learning-based HAR systems become more prevalent, they are often viewed as “black boxes”, offering little insight into the decision-making process. This lack of transparency hampers user trust and accountability, particularly in high-stakes applications such as healthcare or security. To address this, researchers are increasingly focused on developing techniques that improve model interpretability [318], such as attention mechanisms that highlight which features or data points the model focuses on during decision-making, model visualisation tools that provide intuitive insights into neural network behaviour, and explanation methods that allow users to understand the underlying reasons behind specific predictions.

The transferability and generalisability of HAR models remain as significant challenges. Models trained on specific datasets and conditions often struggle to generalise effectively to new environments. Likewise for sensor configurations, unseen activity patterns discovery remains challenging [319]. This limits the deployment of HAR systems in real-world applications. The variability in human actions, bias introduced by sensor placements, and other environmental factors can be vast. To overcome these challenges, techniques like transfer learning and domain adaptation are being explored by HAR researchers. These methods allow HAR models to adapt to new conditions using knowledge from related tasks by artificially generating diverse training samples to simulate real-world variability.

Ultimately, ensuring that HAR systems are user-friendly and meet the requirements of the target audience is achieved by incorporating human-centric design and evaluation [318]. Involving users in the design and evaluation process through participatory design and user-centred evaluation methodologies helps ensure that HAR systems are inclusive, accessible, and capable of addressing real-world challenges effectively [319]. By prioritising human-centric design principles, researchers can create systems that not only perform well in technical terms but are also widely adopted and accepted by users in various domains, from healthcare to smart homes.

These challenges, while significant, offer opportunities for further innovation in the field of HAR. Creating scalable and user-friendly HAR systems that are capable of meeting the demands of real-world applications, HAR researchers can advance the field.

## 8. Conclusions

Human activity recognition (HAR) has surfaced as a crucial field of computer vision and pattern recognition research. Research in this field offers transformative potential across diverse applications across various fields including healthcare, sports, security, and smart environments. HAR technology uses various data modalities like vision sensors, depth sensors, motion capture devices, and wearable sensors. Regardless of sensing medium, the goal of any HAR system is to accurately classify human activities. Each modality comes with its advantages and challenges. For example, RGB-based systems offer rich visual data but are sensitive to environmental factors like lighting. RGB-based sensors have a longer range but are sensitive to occlusions. Depth sensors on the other hand provide reliable 3D spatial information but face limitations in range. Depth sensors are efficient for segmenting the foreground and background but some of the devices can only be used in indoor environments. Motion capture devices have high precision, but they are costly and impractical for everyday use. Wearable sensors are effective for real-time activity recognition in different contexts, but they cause user discomfort over long periods. Moreover, wearable sensors are limited in classifying a limited number of activities.

HAR has been transformed by the transition from conventional rule-based systems to advanced machine learning methods. Traditional methods relied heavily on handcrafted features and pre-defined rules were used. This specific domain knowledge limited their application in recognising simple activities that are inadequate for handling complex action. In contrast, modern machine learning approaches like the deep learning models such as Convolutional Neural Networks (CNNs) and Recurrent Neural Networks (RNNs), have significantly improved the performance of HAR systems. These models automatically extract features and learn from raw data that allow them to adapt to diverse activity contexts and achieve high accuracy.

The pre-processing of data remains a crucial aspect of HAR. Data cleaning, normalisation, and feature engineering are often applied to ensure optimal model performance. Techniques like noise reduction, data augmentation and dimensionality reduction are employed to enhance data quality. Data labelling and segmentation is essential for HAR processes, but these steps remain less explored as they are labour-intensive and error-prone processes. Innovations in semi-supervised and crowdsourced data labelling are helping to reduce these challenges to a certain extent paving the way for efficient data preparation.

Trends in HAR research indicate that multimodal sensor fusion is becoming increasingly important. Hybrid methods integrate diverse sensors to capture comprehensive information about human activities. Edge computing and IoT-based systems allow real-time data processing while reducing latency. Personalised and adaptive models are being developed to cater to individual users’ preferences. The ethical and social implications of HAR technology like privacy concerns, algorithmic fairness, and data security are gaining attention. Researchers emphasise transparency and accountability in the design and deployment of HAR systems.

The practical applications of HAR technology are vast and impactful. In healthcare, HAR systems are used for fall detection, remote patient monitoring, and personalised rehabilitation. Advancements in HAR technology in the medical field can significantly improve patient outcomes and safety. In sports, HAR technology can be used to provide real-time feedback on athletes’ biomechanics, optimising training and reducing injury risks. Security applications benefit from HAR’s ability to detect intrusions and monitor suspicious behaviours, enhancing public safety. Meanwhile, smart environments leverage HAR to automate daily functions, improve energy efficiency, and respond to occupants’ needs in a personalised manner.

Despite these advancements, HAR research continues to face challenges, including variations in human behaviour, model interpretability and achieving reliable recognition in real-world settings. Addressing these issues requires continued innovation and interdisciplinary collaboration to fully realise the potential of HAR technology. Future studies should prioritise strengthening privacy protection with advanced methods such as federated learning to enable decentralised model training without exposing raw data and differential privacy to ensure robust anonymisation in shared datasets. Additionally, multimodal fusion strategies should be further refined, with a focus on adaptive fusion techniques that dynamically adjust sensor weightings based on environmental conditions and task requirements. To enhance real-time processing, research should also explore the development of lightweight deep learning architecture optimised for HAR and leverage edge computing to reduce latency and enhance computational efficiency. Moreover, enhancing model interpretability remains a critical challenge for real-world adoption. Future efforts should also incorporate explainable AI strategies, such as feature attribution methods and attention-based visualisation, to improve transparency and trust in HAR systems. Additionally, benchmarking HAR models across diverse and unstructured environments will be essential to improve their generalisability and robustness.

As HAR research progresses, its integration into smart environments, healthcare, autonomous systems, and security applications will continue to expand, making daily interactions more adaptive, safe, and efficient. This study underscores the significant impact of HAR and highlights the ongoing need for research and development to harness its full capabilities for societal benefit.

## Figures and Tables

**Figure 1 jimaging-11-00091-f001:**
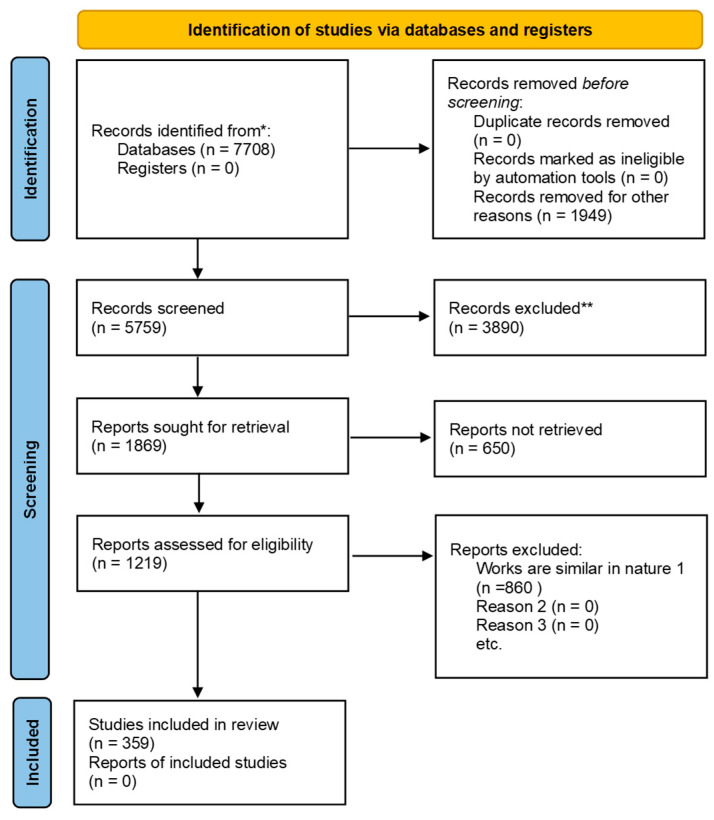
The PRISMA flow diagram used for the systematic identification, screening, eligibility, and inclusion of publications.

**Figure 2 jimaging-11-00091-f002:**
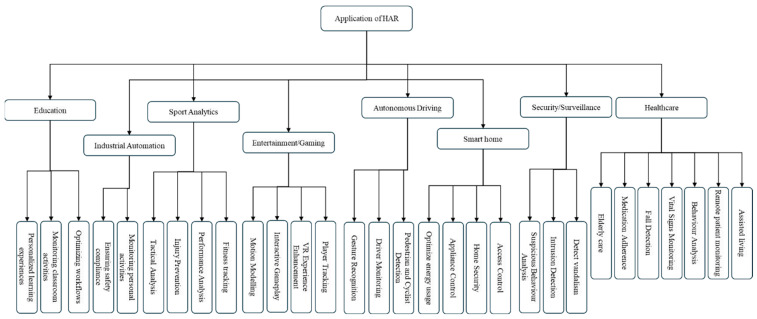
Application of HAR in various fields.

**Figure 3 jimaging-11-00091-f003:**
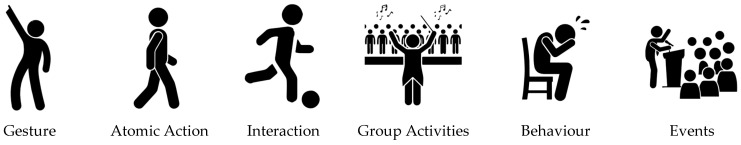
Different type of activities from the literature.

**Figure 4 jimaging-11-00091-f004:**
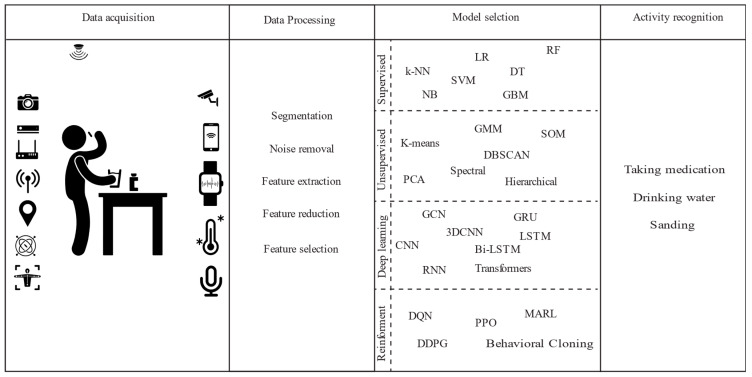
Stages of Human Activity Recognition (HAR).

**Figure 5 jimaging-11-00091-f005:**
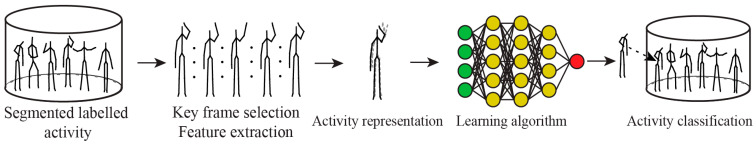
Activity recognition.

**Figure 6 jimaging-11-00091-f006:**
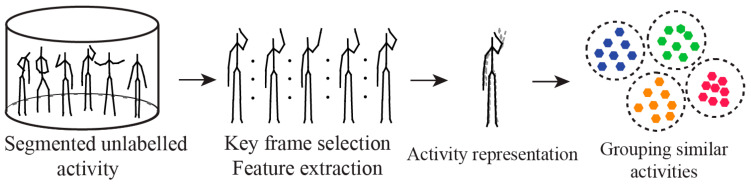
Activity discovery.

**Figure 7 jimaging-11-00091-f007:**
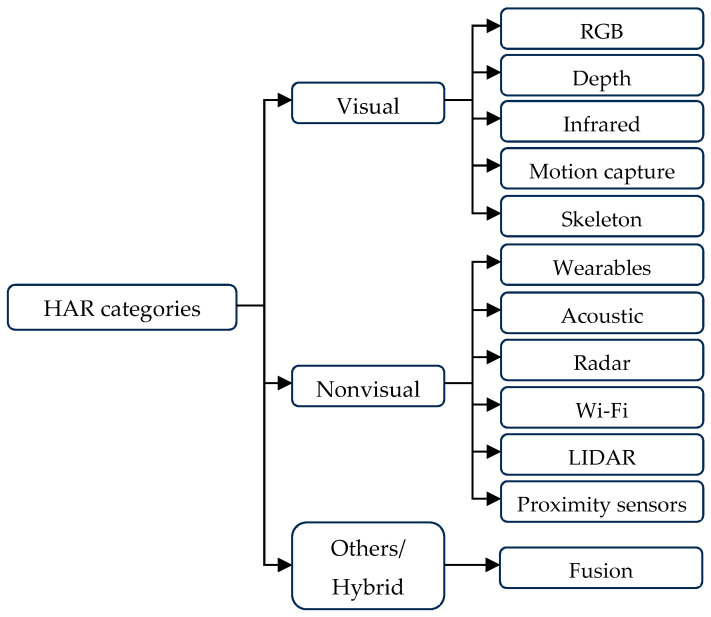
Categories of HAR based on sensor modalities.

**Figure 8 jimaging-11-00091-f008:**
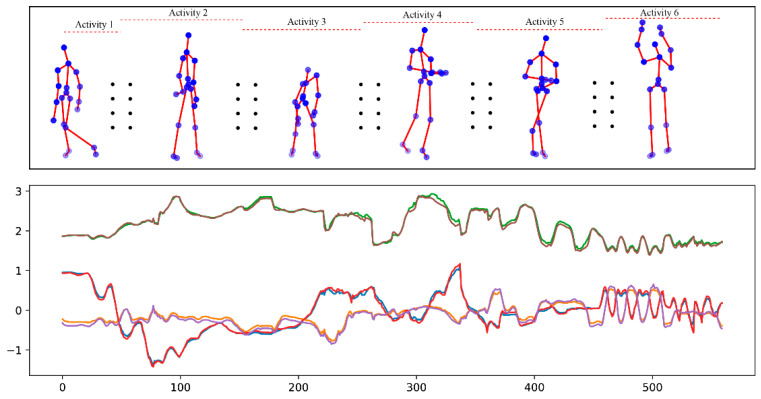
Activities performed in a sequence. Top row depicts skeleton-based representation of multiple activities performed sequentially while the bottom row represents hand joint movements over time.

**Figure 9 jimaging-11-00091-f009:**
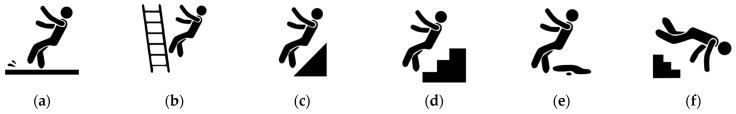
Intraclass variations of fall (**a**) Fall due to imbalanced leg (**b**) Fall from a ladder (**c**) Fall from a slope (**d**) Fall from stairs (**e**) Fall due to slippery surface (**f**) Fall while going downstairs.

**Figure 10 jimaging-11-00091-f010:**
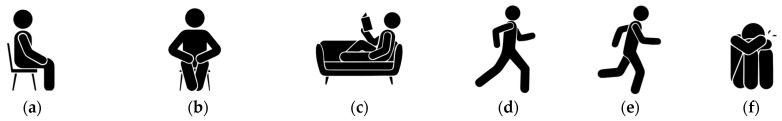
Challenges of vision based HAR (**a**) Sitting viewed from different angles (**b**) Sitting viewed from the front (**c**) Simultaneously sitting and reading (**d**) Running (**e**) Jogging (**f**) Self-occlusion.

**Figure 11 jimaging-11-00091-f011:**
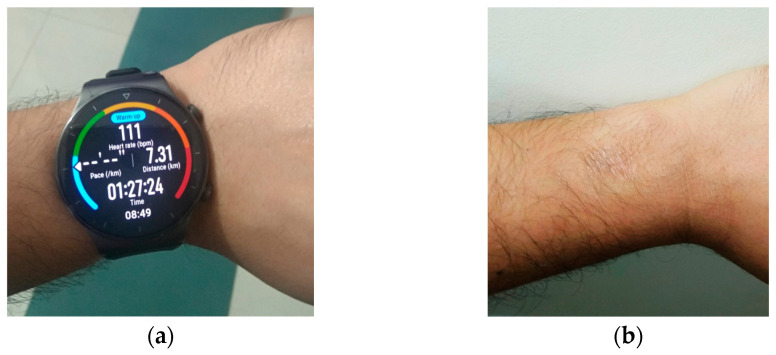
(**a**) Commercially available wearable device attached to user’s wrist to monitor daily exercise. (**b**) Skin irritation caused due to long exposure to wearable devices.

**Figure 12 jimaging-11-00091-f012:**
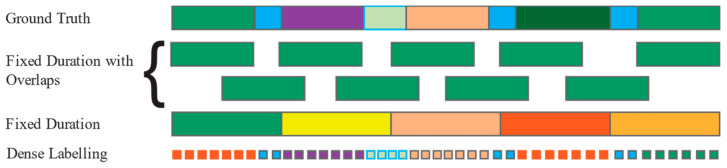
Existing windowing methods commonly applied for segmenting human activity data.

**Table 1 jimaging-11-00091-t001:** Unsupervised activity discovery methods.

Methods	Year	Modality	Top Accuracy	Datasets Used
Estimation-based [77]	2008	Wearable	72.7	[77]
UnADevs using temporal behaviour assumption [73]	2017	Wearable	52, 61.0	JSI-ADL, REALDISP
Stream sequence mining [79,80]	2014	Smart Home	-	CASAS
Discover unknown activities [81]	2016	Smart Home	68.01	CASAS
Incremental clustering using K-means [82]	2013	Skeleton	71.9	CAAD60
Particle swarm [83]	2023	Skeleton	80.1, 58.10, 64.1, 45.1, 40.1	CAD-60, UTK, F3D, KARD, MSR
Long-term visual behaviour [84]	2015	RGB	74.75	Gaze [84]
Unsupervised discovery using spectral clustering [85]	2006	RGB	50.01, 52.1, 49.5	Figure Skating [85], Baseball, Basketball
HMDSM [86]	2019	Smart Home	91.5	UASH [86]
Topic models [87]	2014	Smart Home	85.1	[87]
Evaluation of multiple clustering algorithms [76] K-means, Spectral, GMM,	2023	Skeletons	70.1, 68.05, 74.5	CAD60, UTK, UBDKinect
DBSCAN [68]	2014	Smart phone	80.4	[68]

**Table 2 jimaging-11-00091-t002:** Various HAR modalities from literature with summarised with pros and cons.

Modality	Devices	Sample	Pros	Cons
Silhouette [88]	Different Cameras	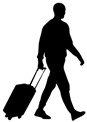	Provides basic humanoid figure and posture information Computationally efficient Clear representation of human motion	Lacks detailed texture and object information Sensitive to occlusions and background noise Challenges in handling variability
RGB	Standard RGB cameras	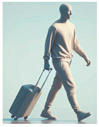	Rich visuals with detailed features Widely available dataset and sensing mediums Can be directly fed to deep learning methods for rich feature learning and scene analysis	Sensitive to occlusions and background noise and change of view Limited spatial and temporal resolution Invasive to privacy
Infrared [89]	Infrared Cameras	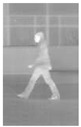	Suitable for low dark/light environments Less intrusive in terms of privacy compared to visible light cameras Can penetrate through certain materials and surfaces that may hinder visible light such as fog	Sensitive to temperature and daylight exposure Lacks detailed texture and object information Costlier than visible light cameras
Depth [90]	Depth Cameras such as Microsoft Kinect/Kinect Azure/	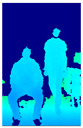	Privacy preserving and provides rich 3D structural information. Efficient for separating background and foreground objects Invariant to changes in lighting conditions and partial occlusion	Sensitive to daylight Limited range and may produce noisy data, at longer distances. Lacks detailed texture and object information
Skeletons [34]	Derived Sensors/RGB Cameras	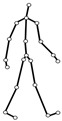	Representing human motion using a simplified skeletal model Invariant to background clutter and variations in lighting conditions and change of view Captures the underlying structure of human motion, including joint movements and temporal sequences	Heavily reliant on accurate estimation of joint positions Lacks texture, scene, and object information Skeleton-based methods may struggle to handle missing or inaccurate joint detections
Mocap [91]	Commercial Mocap systems	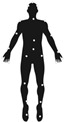	Provides highly accurate and detailed motion data Capturing the movement of individual body parts with precision Action recognition can be used for animation and visualizations of human movement	Mocap devices are expensive and costly to maintain Requires individuals to attach markers to the body Multiple sensors are required for capturing data, mostly suitable for controlled environments
Point cloud [92]	Depth sensors (like Kinect, LiDAR, range sensors), stereo cameras	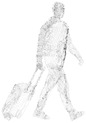	Captures the 3D spatial information of the scene and subject Robust to occlusions caused by obstacles Point cloud data can be generated from LiDAR sensors or depth sensors	May exhibit data sparsity and irregularity Processing and analysing point cloud is computationally intensive Sensitive to calibration and environmental disturbances
Event stream [93]	Specialised event cameras	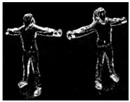	High temporal resolutions with reduced visual redundancy Longer viewing and tracking range and insensitive to motion blur Data are generated only when there is a change in the scene	Tracking devices are expensive and costly to maintain Difficult to model activities with less motions Multiple sensors are required for capturing data, suitable for controlled environments
Thermal [94]	Thermal Cameras	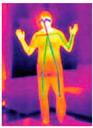	Suitable for low dark/light environments and able to penetrate through certain surfaces Less intrusive in terms of privacy compared to visible light cameras Can detect anomalies in human body temperature, such as fever	Provides lower spatial resolution compared to visible light cameras and sensitive to temperature difference Lacks detailed texture and object information Costlier than visible light cameras
Audio [95]	Microphones, audio recording devices	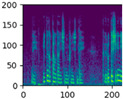	Offers non-intrusive monitoring of human activities Provides complementary information to visual and motion-based sensing modalities Can be applied in a wide range of environments and contexts	Lacks the discriminative power required to distinguish between different activities May raise privacy concerns Susceptible to environmental factors such as background noise, and acoustic interference
Radar [96]	Radar devices	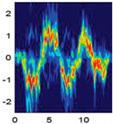	Can penetrate through obstacles such as walls and furniture Less affected by environmental conditions such as lighting, weather, and ambient noise High spatial and temporal resolution, capturing detailed information about the position	Expensive and complex to deploy compared to other sensing modalities Lacks the discriminative power required to distinguish between different activities May raise privacy concerns
WiFi CSI [97]	Standard WiFi routers, network interface cards	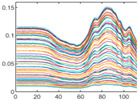	Non-intrusive method for monitoring of human activities Provides fine-grained information about the propagation of wireless signals Can be used in ubiquitous indoor environments	Limited range and coverage Sensitive to environmental changes such as multipath reflections, obstacle locations May raise privacy concerns
Inertial sensors [98]	Smartwatches, fitness trackers, smartphones with IMUs (accelerometers, gyroscopes, magnetometers)	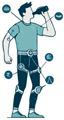	Sensors are compact and lightweight, provided multimodal data such as speed and angle Low-cost devices, fine grained signals, privacy protecting Can capture motion data in real-time, allowing for immediate feedback and analysis of human activities.	Lacks contextual information about the environment or activity context Signals are noisy, multiple sensors are required for high label activity recognition Need to attach to body and cause user discomfort
Environmental sensor [39]	Sensors embedded in floors or wearables; various proximity sensor devices	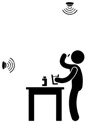	Consume low power compared to other sensing modalities Non-intrusive monitoring of human activities Provide contextual information about the surrounding environment	Limited discriminative power for distinguishing activities Performance is affected by obstacles and other moving objects Requires cautious planning and installation to ensure optimal sensor placement and coverage

**Table 3 jimaging-11-00091-t003:** Summary of key publicly available HAR datasets.

Dataset	Year	Modality	#Class	#Subject	#Sample	#Viewpoint
UCF101 [99]	2012	RGB	101	-	13,320	-
Sports-1M [100]	2014	RGB	487	-	1,113,158	-
ActivityNet [101]	2015	RGB	203	-	27,801	-
THUMOS Challenge 15 [102]	2015	RGB	101	-	24,017	-
Charades [103]	2016	RGB	157	267	9848	-
InfAR [104]	2016	IR	12	40	600	2
NTU RGB+D RGB+D [105]	2016	RGB, S, D, IR	60	40	56,880	80
DvsGesture [106]	2017	Event Stream	17	29	-	-
FCVID [107]	2017	RGB	239	-	91,233	-
Kinetics-400 [108]	2017	RGB	400	-	306,245	-
PKU-MMD [109]	2017	RGB, S, D, IR	51	66	1076	3
Something-Something-v1 [110]	2017	RGB	174	-	108,499	-
Kinetics-600 [111]	2018	RGB	600	-	495,547	-
RGB-D Varying-view [112]	2018	RGB, S, D	40	118	25,600	8 + 1 (360°)
DHP19 [94]	2019	ES	S	33	17	-
Drive&Act [113]	2019	RGB, S, D, IR	83	15	-	6
Egogesture [114]	2018	RGB, D	83	50	2,953,224	
Moments in time [115]	2019	RGB	339	-	∼1,000,000	-
NTU RGB+D 120 [116]	2019	RGB, S, D, IR	120	106	114,480	155
RareAct [117]	2020	RGB	122	-	905	-
MoVi [118]	2021	RGB, Mocap	21	90	7,344,000+	4
UAV-Human [119]	2021	RGB, S, D, IR, etc.	155	119	67,428	-
HOMAGE [120]	2021	RGB, IR, Au, Ac, Gy	86		26,000+	5
Ego4D [121]	2022	RGB, Au, Ac, etc.	-	923	-	Egocentric
EPIC-KITCHENS-100 [122]	2022	RGB, Au, Ac	-	45	89,979	Egocentric
EPIC-KITCHENS-55 [123]	2023	RGB	-	397	2,000,000+	Egocentric

Table keys, Gy = Gyroscope, S = Skeleton, D = Depth, Ir = Infrared, Au = Audio, Ac = Acceleration, Ps = Pressure.

## Data Availability

Data are contained within the article or Appendix A.

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
