# Peer review of "Machine Learning for Human Activity Recognition: State-of-the-Art Techniques and Emerging Trends"

_2313-433X, 2025, doi:10.3390/jimaging11030091_

Round 1
Reviewer 1 Report
Comments and Suggestions for Authors
The paper is a -very long- dissertation about: review of human action recognition (HAR) systems, types of human actions, sensing/tracking technologies related to HAR, processing methods related to HAR, pros and cons. The paper presents a detailed listing and description of the items belonging to such topics, and it identifies the open challenges in the field of HAR applications.
-The major issue of the presented review is that it is missing: search criteria, inclusion/exclusion criteria, explanation of analysis of results and other information proper to a literature review investigation process. Authors are kindly encouraged to provide these information in detail; authors are also invited to consider the use of PRISMA protocol.
-Authors are encouraged to consider to also add a structured summary/table depicting available real devices for sensing/tracking humans.
-Paragraph 4.3.1: for all the mentioned equations, please specify the source.
-In Tables 1, 2, 10, 11, 12, 13, 14, 15 the reader cannot clearly realize if the mentioned works (first column) are seminal/early works or are other works. The authors are encouraged to properly specify/clarify when:
A) the sources in the first column of the table are the "original" work that has produced
the entry (the line of the table);
B) the sources in the first column of the table are just works where the entry (mentioned in the line of the table) is used.
-In the text there are various strings like: "Error! Reference source not found", (for example, line 367); plase fix them.
-Figure 11: the figure has truncated words, please rearrange the figure.
Reviewer 2 Report
Comments and Suggestions for Authors
This review describes the current research status of HAR. This paper covers almost all data modes including images, point clouds, ranging sensors, RGB, depth, skeleton, infrared sequences, event streams, motion capture devices (Mocap), acoustic sensors, wearable devices, radar, Wi-Fi and sensor fusion. The main text of the article is over 60 pages long. This is the longest review of the main text I have ever read. The authors discuss the definition of HAR, the data model, the public data set, and the differences between different methods, and make a summary and prospect.
There are no major problems in the content except for a few typesetting or grammar errors (eg. on Page 10 there is an error that is 'Error! Reference source not found.' ).
Besides, the authors need to ask whether the editor is allowed to publish nearly 100 pages of the paper. If it is necessary to reduce the pages, the authors should not introduce the methods in too much details, but should highlight the characteristics and differences between the methods.
Reviewer 3 Report
Comments and Suggestions for Authors
I think the article is clearly too long. It should be shortened by at least half, focusing on a few clearly presented examples that the authors have already dealt with in the past. Such an approach would provide more evidence that the authors have sufficient competence in the field of HAR.
---------------------
The introduction should be rewritten. Instead of listing numerous applications of Human Activity Recognition (HAR) with extensive citations, it would be more effective to begin with a specific example from the authors' own research, if possible, and delve into its analysis. While the breadth of HAR applications (healthcare, smart environments, sports, etc.) is important context, the current introduction relies too heavily on a long list of citations rather than a focused narrative. A more compelling approach would be to showcase the significance of HAR through a concrete example and then broaden the discussion to encompass the various modalities and machine learning techniques employed in the field. This would create a more engaging and less citation-heavy introduction.
------------
Additionally, the paper currently reads more like a list of references rather than an in-depth review. With 517 references, it feels more like a semi-automatically generated compilation rather than a review paper that allows readers to delve into and learn something substantial about HAR.
----
The second chapter could benefit from being more concise and focused. It currently gives the impression that the authors are attempting to encompass the entire history and various methods of machine learning within the context of the HAR field.
----
The 'Data Collection and Pre-processing' section (chapter 3), while aiming to be comprehensive, dedicates five pages to detailed tables of various HAR datasets. This approach risks detracting from the core HAR review ideas. The sheer volume of tabular data might discourage readers from engaging with the section's key insights.
I strongly suggest a more focused approach. Instead of presenting an exhaustive list, consider highlighting a select few of the most influential and widely used datasets. These datasets should be reviewed in depth, emphasizing their unique characteristics, strengths, and potential applications within the HAR field. This would provide readers with a more digestible and informative overview, allowing them to truly learn from the presented data.
The full dataset tables, while valuable, would be better suited for an appendix or a separate, supplementary document with some Internet (i.e. GitHub) link. This would allow readers who require comprehensive data to access it, without interrupting the flow of the main article.
Comments on the Quality of English LanguageThe quality of English in the manuscript is generally acceptable. However, there are areas where improvements could enhance clarity and readability.
Reviewer 4 Report
Comments and Suggestions for Authors
This manuscript broadly covers the latest technologies and trends in the field of human activity recognition (HAR). It provides a comprehensive overview, addressing sensor data, deep learning methods, multimodal fusion, and other key aspects. However, there are several issues that need to be revised. My suggestions are as follows:
- Some sections contain significant redundancy. For example, Sections 2.1 and 2.5 both discuss activity classification, while Section 2.6, which discusses sensors, overlaps with previous sections. It is recommended to merge and optimize redundant content.
- Some terminology is inconsistent. For example, "RGB-based HAR" and "Vision-based HAR" are used interchangeably. It is recommended to standardize the terminology, such as using "Vision Sensor HAR" consistently.
- When discussing deep learning methods, only CNN, RNN, and Transformer are listed, but there is a lack of in-depth analysis of why these models are suitable for HAR. The section on multimodal sensor fusion merely lists different methods without discussing specific fusion strategies (e.g., early fusion, late fusion, hybrid fusion).
- The conclusion mentions future challenges (such as real-time processing, multimodal fusion, and privacy issues) but does not provide specific research recommendations.
Round 2
Reviewer 1 Report
Comments and Suggestions for Authors
The paper is a -very long- dissertation about: review of human action recognition (HAR) systems, types of human actions, sensing/tracking technologies related to HAR, processing methods related to HAR, pros and cons. The paper presents a detailed listing and description of the items belonging to such topics, and it identifies the open challenges in the field of HAR applications.
PRISMA protocol has been used (with related intermediate and final steps, results and descriptions).
-In this second version of manuscript, authors are asked to fix the following points:
1) Equations are missing their related source reference. The reader would appreciate to be able to know what is the source of equations, even when it is the case of "well-known" equations.
2) In the Supplementary Material: the text of Paragraph 3 (that is about Tables S8-S13) is the same text of Paragraph 2.
Reviewer 3 Report
Comments and Suggestions for Authors
The authors have addressed the previous comments. While further refinements based on the initial review would strengthen the paper, the current version is suitable for publication.
Reviewer 4 Report
Comments and Suggestions for Authors
The authors have revised it well and it's ready for publication now.
